# Skin properties and afferent density in the deterioration of tactile spatial acuity with age

Davide Deflorio, Massimiliano Di Luca and Alan M. Wing

*School of Psychology University of Birmingham, Birmingham, UK*

Handling Editors: Richard Carson & Vaughan Macefield

The peer review history is available in the Supporting information section of this article (https://doi.org/10.1113/JP283174#support-information-section).

**Abstract** Tactile sensitivity is affected by age, as shown by the deterioration of spatial acuity assessed with the two-point discrimination task. This is assumed to be partly a result of age-related changes of the peripheral somatosensory system. In particular, in the elderly, the density of mechano-receptive afferents decreases with age and the skin tends to become drier, less elastic and less stiff. To assess to what degree mechanoreceptor density, skin hydration, elasticity and stiffness can account

**Davide Deflorio** is a PhD candidate at University of Birmingham, Birmingham, UK. Using psychophysics, computational modelling, measurements of finger properties and analysis of contact dynamics, he is investigating the sensory mechanisms underlying tactile perception. His goal is to shed light on the basic sensory mechanisms involved in different aspects of touch including passive and active exploration of an object's features and the changes that occur throughout the lifespan.

for the deterioration of tactile spatial sensitivity observed in the elderly, several approaches were combined, including psychophysics, measurements of finger properties, modelling and simulation of the response of first-order tactile neurons. Psychophysics confirmed that the Elderly group has lower tactile acuity than the Young group. Correlation and commonality analysis showed that age was the most important factor in explaining decreases in behavioural performance. Biological elasticity, hydration and finger pad area were also involved. These results were consistent with the outcome of simulations showing that lower afferent density and lower Young's modulus (i.e. lower stiffness) negatively affected the tactile encoding of stimulus information. Simulations revealed that these changes resulted in a lower build-up of task-relevant stimulus information. Importantly, the reduction in discrimination performance with age in the simulation was less than that observed in the psychophysical testing, indicating that there are additional peripheral as well as central factors responsible for age-related changes in tactile discrimination.

(Received 27 October 2022; accepted after revision 13 December 2022; first published online 19 December 2022)
**Corresponding author** A. M. Wing: School of Psychology University of Birmingham, Birmingham B15 2TT, UK. Email: a.m.wing@bham.ac.uk

**Abstract figure legend** We combined measurements of finger properties (1), psychophysics (2) and simulation of first-order tactile neurons (3) to investigate the contribution of peripheral factors to the deterioration of tactile spatial acuity with age. We measured skin biological elasticity, hydration and finger pad area in Young and Elderly participants to predict the behavioural outcome on the psychophysical task. In addition, we simulated the response of first-order tactile neurons [SA1 (i.e. type 1 slowly adapting fibres) and RA1 (i.e. type 1 rapidly adapting fibres) afferents] to determine whether changes in Young's modulus, related to biological elasticity, and afferent density, related to finger pad area, affect the informativeness of the peripheral neural response.

### Key points

- Ageing effects on tactile perception involve the deterioration of spatial sensitivity, although the contribution of central and peripheral factors is not clear.
- We combined psychophysics, measurements of finger properties, modelling and simulation of the response of first-order tactile neurons to investigate to what extent skin elasticity, stiffness, hydration, finger pad area and afferent density can account for the lower spatial sensitivity observed in the elderly.
- Correlation and commonality analysis revealed that age was the most important factor to predict behavioural performance. Skin biological elasticity, hydration and finger pad area contributed to a lesser extent.
- The simulation of first-order tactile neuron responses indicated that reduction in afferent density plays a major role in the deterioration of tactile spatial acuity.
- Simulations also showed that lower skin stiffness and lower afferent density affect the build-up of stimulus information and the response of SA1 (i.e. type 1 slowly adapting fibres) and RA1 (i.e. type 1 rapidly adapting fibres) afferent fibres.

## Introduction

Ageing is characterised by progressive impairments of vision, hearing, taste and smell (Peelle, 2020). The sense of touch is no exception in this decline because it is affected by age in several ways, including decreasing pressure sensitivity (Bowden & McNulty, 2013), vibrotactile sensitivity (Gescheider et al., 1994) and spatial acuity. The latter has been extensively studied in a variety of tasks. Stevens and Cruz (1996) showed higher thresholds for elderly participants than their younger counter-parts (mean age 77.5 *vs.* 22.7 years) in three different tasks: two-point gap detection, line orientation and line length discrimination. Similarly, Goldreich and Kanics (2003) observed a linear decrease of sensitivity in a grating orientation discrimination task with age for both sighted and blind participants ranging from 19.7 to 71.6 years. Furthermore, age-related impairment of tactile spatial acuity has also been reported for the two-point discrimination paradigm (Bowden & McNulty, 2013; Woodward, 1993). This deterioration appears to be more

evident at the toes and fingers, as shown by Stevens and Choo (1996) who tested the sensitivity of different body parts across participants between 8 and 87 years of age by means of the gap detection task.

There are at least three age-related anatomical and morphological changes that affect the glabrous skin of the finger that might contribute to a decline in tactile performance. First, with age, the skin tends to lose its biological elasticity (i.e. the ability to recover its original shape following deformation) and, for some individuals, it becomes less stiff (i.e. it shows an increased displacement for a given indentation force) (Boyer et al., 2009). Yang et al. (2018) used real-time shear wave elastography to measure Young's modulus (YM) in participants from three different age groups. YM determines the stiffness of the initial indentation of the skin, that is, when the force applied is small and the amount of indentation is *quasi* linearly-proportional to the force applied. Stiffness was lowest (low YM, which means softer skin) in participants aged 3−19 years (mean 25.2 kPa). Stiffness was highest in the age group of 20−50 years (mean 36.1 kPa) and then it decreased for and older participants >50 years old (mean 29.6 kPa), indicating a softening of the skin. Second, ageing glabrous skin becomes drier, as observed by Skedung et al. (2018) who showed that skin hydration was significantly lower in elderly compared to young participants, age (mean ± SD) 73 ± 4.5 *vs.* 22 ± 1.5 years. Third, mechanoafferents exhibit significant change with age. Myelination of axons deteriorates (Peters, 2002), size and morphology of neurons change, and the number of some of the mechanoreceptors is reduced (García-Piqueras et al., 2019). García-Piqueras et al. (2019) employed immunohistochemistry and immuno-fluorescence to measure the number of Merkel, Meissner and Pacinian (PC) receptors on post-mortem samples in three age-groups (20−39, 40−59 and 60−90 years). The results showed that the number of Merkel and Meissner cells was four to five times lower in the 60−90 years age-group compared to the 20−39 years age-group. In addition, the size of receptive organs was reduced and morphology was compromised. However, no difference in terms of number, size and morphology was observed for the PC corpuscles between the three age groups. Importantly, the age-related effect of lower number of receptors might be compounded by increases in finger width (Dequeker & Vadakethala, 1979), resulting in reductions in receptor density with age.

These skin properties determine how a tactile stimulus activates the population of mechanoreceptors and probably affects the encoding of the sensory information that is transmitted downstream. However, there is little evidence that links these changes to the decline of tactile perception. The present study combines psychophysics of tactile discrimination (two-point threshold) and measurements of finger properties to investigate the degree to which peripheral factors (biological elasticity, hydration, finger pad area) might drive the deterioration of tactile spatial sensitivity observed in the elderly in the index finger.

To interpret the behavioural findings, the spiking activity of a population of type 1 slowly adapting fibres (SA1) and type 1 rapidly adapting fibres (RA1) was simulated in response to stimuli modelled on those employed in the behavioural experiment. The simulation was based on the model of Saal et al. (2017), which allows the creation of a flexible representation of afferent fibres (i.e. density, location) and implements a simplified basic skin mechanics that can be manipulated in terms of YM to account for differences in skin properties.

The focus was on determining whether the spatial layout of stimuli applied to the skin can be decoded based solely on the firing rate variations of the virtual neurons (i.e. rate coding) and how the decoding performance is affected by lower YM (Yang et al., 2018) and lower mechanoafferent density (García-Piqueras et al., 2019). A spatial code was chosen because it was previously shown to be a viable mechanism through which stimulus shape, or stimulus spatial layout, can be extracted under naturalistic conditions in the presence of noise (e.g. shift in location of the indentation). Other potential mechanisms subserving the extraction of shape information, such as first spike latency (Johansson & Birznieks, 2004), are considered in the Discussion.

Model simulations were run separately for Young and Elderly groups with the goal of predicting behavioural performance. The two virtual groups differed in terms of skin stiffness and mechanoreceptor density. The simulated neurophysiological data were also used to examine how stimulus information unfolds over time in the two age-groups and which afferent type might be more closely tuned to the fine spatial details of the stimulus employed.

## Methods

### Ethical approval

The study was approved by the STEM Ethical Committee at University of Birmingham (ERN_09-528AP24) and conformed to the standards set by the *Declaration of Helsinki*, apart from registration in a database. All participants gave their written informed consent before the beginning of the experiment. Both Young and Elderly participants were reimbursed for taking part in the experiment.

### Participants

Fourteen young participants (nine females, age range 18−32 years, mean ± SD age = 23.85 ± 3.25 years) and fourteen elderly participants (six female, age range 60−86

years, mean $\pm$ SD age $=72.36 \pm 8.39$ years) were recruited for this study. There were 13 right-handed participants in the Young group and 11 in the Elderly group. One of the study investigators (DD) was tested and the data were included in the Young group. Eligibility criteria for both groups were normal or corrected-to-normal vision, independence in activities of daily living, absence of physical hand injury, and absence of motor and sensory impairment as a result of arthritis or other causes (e.g. carpal tunnel syndrome, diabetes) based on self-report. Furthermore, participants confirmed they were not taking any medication with central nervous system effects.

Testing of participants involved first measuring skin elasticity, hydration and fingertip size, in this specific order, followed by psychophysical testing. All testing was carried out with the index finger of the right hand because previous studies showed no difference in sensitivity between hands for the two-point discrimination task (Kalisch et al., 2009).

### Psychophysical task and stimulation setup

Blindfolded participants were tested on a spatial two-point discrimination task to determine sensitivity thresholds for each individual. The threshold was defined as the two-point distance at which participants could respond correctly on 75% of trials and here referred to as the Just Noticeable Difference (JND). We used a 2IFC procedure (i.e. two intervals forced choice), in which a single flat-ended brass pin and a pair of brass pins were moved, first one then the other, vertically down onto the supported index finger pad in random order. The finger was positioned, finger pad facing up, at the beginning of the experiment so that the pins made contact with the central portion of the finger pad and did not touch the outer edges or the tip of the finger. The position was monitored throughout the experiment and re-adjusted when needed (e.g. after a break or in case of any small movement of the hand).

Participants were instructed to verbally report which interval contained the two-point stimulus (i.e. first or second) and to give their best guess when they were not sure about the answer. We tested seven different separation levels for the pair of pins which differed between the two age-group based on a pilot study. For the Young group, the separations were 0.1, 0.3, 0.6, 1, 1.3, 1.6 and 2 mm. For the Elderly group, the separations were 1, 1.5, 2, 2.5, 3, 3.5 and 4 mm. Each separation distance was presented 10 times for a total of 70 trials. To control the contact area, we chose the single pin to have a diameter of 0.6 mm and each of the pins in the pair a diameter of 0.4 mm. The resulting contact area was 0.28 mm$^2$ for the single and 0.25 mm$^2$ for the two pins together. All the pins were levelled and inspected with microscope and micrometre to ensure they all had the same length.

The pins used as tactile stimuli were attached to a stepper motor to automatically and accurately control the separation distance through Arduino custom software. The single pin was fixed on one end of the stepper motor. Similarly, one pin of the pair was attached to the other end, and the other pin of the pair was attached to a screw-and-slider actuator (Fig. 1).

On each trial, with the participant's right index finger rested on a support (Fig. 1), the finger pad was passively stimulated by either one or two pins. To control the timing, force and location of the application of the tactile stimulus, and especially the velocity of indentation which has been shown to affect discrimination in this kind of task (Yokota et al. 2020), we employed a custom-built apparatus based on a robotic device (Force Dimension Delta 3; Force Dimension, Nyon, Switzerland). At each trial, the Delta lowered and raised the tactile stimulus on the participant's finger. The payload of the Delta comprised the single pin and two pins in the pair whose distance was controlled by mounting one of the pins on a slider so that its position could be adjustable by a stepper motor. The Delta delivered either the single or the two pins to the participant's finger by moving sideways and lowering on a random location of the participant's fingertip. The force applied by the pins on the finger was controlled using an ATI Nano17 force sensor (ATI Industrial Automation, Apex, NC, USA). The Delta was programmed to move and indent the skin at 4 mm/s. The force sensor was calibrated to account for gravity and the marginal effect of acceleration on the force sensor readings. The target force level was set to 0.25 N. The resulting indentation depth ranged between 1 and 2.5 mm. The duration of each indentation was ∼3 s from the initial contact to the release. The interval between the first and second stimulus was ∼5 s because of hardware repositioning constraints and to allow the skin to fully recover between the first and second contact.

### Finger measurements

Skin elasticity and hydration measurements were carried out with a Cutometer dual MPA 580 (Courage and Khazaka Electronic GmbH, Cologne, Germany). Elasticity was measured with the aspiration probe, which was placed in contact with the right index finger resting on the table. The probe was set to apply a negative pressure of 450 mbar and to record the skin deformation during the aspiration and the release phase. Each measurement consisted of three suctions with on-time of 2 s, off-time of 5 s, for a total of 21 s. Biological elasticity (Ur/Uf) was computed using Cutometer software (EnviroDerm, Longhope, UK) as the ratio between the amount of immediate skin retraction (Ur) during the release phase and the maximum elongation of the skin during the aspiration phase (Uf).

The hydration level of the stratum corneum was measured with a Corneometer CM 825 (Courage and Khazaka Electronic GmbH), which expresses changes in water content in arbitrary units. We repeated the hydration measurement five times and computed the average value. Finger pad size was manually measured with a digital calliper having a resolution of 10 $\mu$m. We measured the length and width of the index finger pad and calculated its area (finger pad area = width $\times$ length). The length was measured by aligning one tip of the calliper with the joint connecting the distal phalanges with the intermediate phalanges, and the other tip of the calliper with the tip of the finger. The width was measured by aligning the two tips of the calliper on the left and right side of the finger pad.

### Overview of the TouchSim model

To simulate the response of the tactile fibres that innervate the tip of the index finger, an existing model (TouchSim) (Saal et al., 2017) was used to generate our specific stimuli and manipulate the parameters characterising the two groups (see below: Simulation of neurophysiological data).

In the model by Saal et al. (2017), several assumptions were made to create an efficient framework suitable for real-time applications. Saal et al. (2017) modelled the virtual skin with continuum mechanics to represent the stresses acting on the receptors that serve as input to a leaky integrate-and-fire neural model. Thus, the skin is assumed to be flat, homogenous and elastic, with isotropic behaviour and does not include any hard structures (e.g. bone), nor fingerprints. We argue that these assumptions

are not a major concern in the present work for the reasons outlined in the Discussion.

Importantly, this model allows direct manipulation of several parameters, such as the density and distribution of the virtual mechanoafferents, the YM of the skin, and the location and depth of the indentation. Being able to assess the effects of these parameters on the response of afferent population makes it possible to address open questions related to aging. In particular, we employed this model aiming to understand whether lower YM and lower afferent density, as observed in the elderly population, have an impact on the very first stage of tactile perception.

In the present study, the simulated population response in the Elderly group depends exclusively on the stiffness of the virtual skin (i.e. YM) and the number of mechanoreceptive afferents, whereas the effects of altered morphology and size of the receptors are not included. This approach was aimed at gauging the extent to which changes in skin properties such as lower YM and lower mechanoafferent density limit the amount of information in the spatial activation of mechanoafferent units, assuming no other difference in the response properties of first-order neurons between young and elderly.

### Simulation of neurophysiological data

Virtual Young and Elderly groups were defined based on the YM of the virtual skin and the afferent density. We set the YM to 50 kPa for the Young group (default value in the original model) and to 35 kPa for the Elderly group. The choice of a lower value for the Elderly group was informed by the findings of Yang et al. (2018) and Boyer et al. (2009). In particular, Boyer et al. (2009) showed that biological

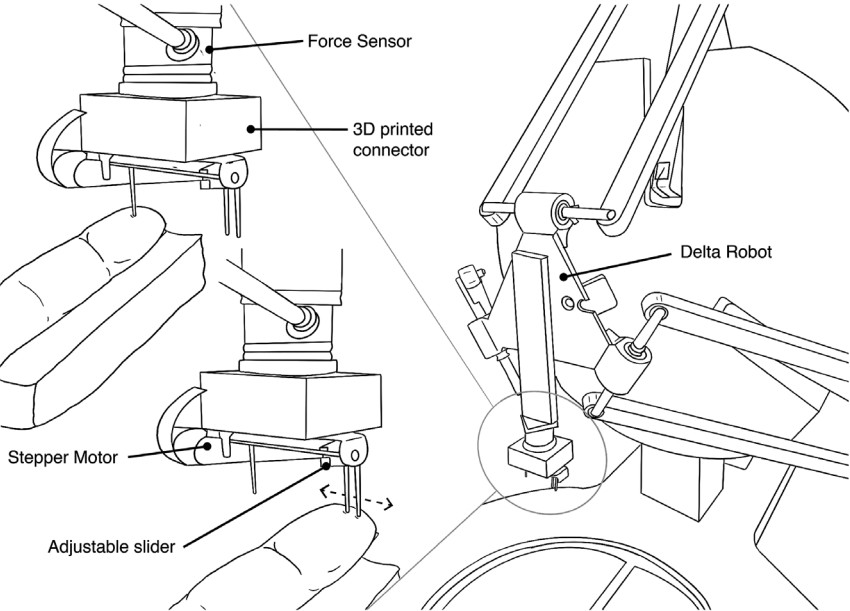

**Figure 1. Schematic view of the stimulation setup**
Line drawing of the experimental setup used for the passive presentation of stimuli for the two-point discrimination task. Right: Force Dimension Delta 3 robot. Left: details of the robotic end effector. The drawing shows the force sensors attached to a custom-made 3D printed part used to mount the stepper motor on the force sensor. The distance between the two pins was controlled with an adjustable slider.

elasticity as measured with the Cutometer (Ur/Uf) and skin stiffness as measured with an indentation device both decrease with age and show a significant positive correlation. In the model by Saal et al. (2017), YM is used to define the stiffness of the skin, such that lower values will determine a larger deformation for the same indenting force.

In both the virtual groups, the simulations included the response of SA1 and RA1 fibres, which are known to have small receptive fields and are both potentially involved in fine spatial sensitivity. Type 2 rapidly adapting afferents (PC) were not included because their low density and large receptive fields mean that they are not tuned to fine spatial details, as previously shown by Saal et al. 2017. Type 2 slowly adapting fibres (SA2) were not included because they are not available in the model.

Virtual afferents were arranged on a square grid covering an area 1 cm$^2$ around the centre of the virtual index fingertip. To make the afferent distribution more biologically relevant, $x$ and $y$ co-ordinates of each fibre were jittered by adding a random value taken from a normal distribution with mean = 0 and SD = afferent spacing × 1/5 (Fig. 2). The afferent spacing was 1.1 mm for SA1 and 0.8 mm for RA1 units in the Young group, and 1.2 for SA1 and 1.8 for RA1 units in the elderly group. For the Young group, the modelled density of virtual afferents was in line with the observations of Johansson and Vallbo (1979), resulting in 121 SA1 units and 196 RA units. For the Elderly group, the number of SA1 and RA1 was 49 and 100, respectively (Fig. 2). This choice was informed by the work of García-Piqueras et al. (2019) who observed a progressive reduction of receptors in the finger pad across the lifespan.

The set of simulated stimuli applied to the fingertip reproduced the pins used in the behavioural experiment (i.e. single pin *vs*. pair of pins) in terms of contact area and separation distance. Importantly, the set of separation distances was the same in the two simulations (i.e. from 0.1 to 2 mm for both). This was carried out to determine the impact of our manipulations over the same range of stimuli and to facilitate the comparison between Young and Elderly groups.

The virtual stimuli were orthogonally indented into the centre of the modelled index finger. To make the simulated conditions similar to the actual indentations, a trial-to-trial jittering of the position of the virtual stimulus was added to mimic small variations in the actual stimulus presentations as a result of small movements of the finger during the psychophysical experiment. Forty different positions were generated by applying a random value in the two directions obtained from a normal distribution with mean = 0 and SD = 0.5 mm.

The indentation pattern was simulated as a ramp-and-hold function with a ramp-on phase of 100 ms, a sustained hold of 300 ms and a ramp-off phase of 100 ms. Such timings result in a shorter duration than the one used in the human experiment, although it was necessary to not saturate the probability of a correct response in some of the conditions. We used five indentation depths ranging from 1 mm to 2.5 mm in steps of 0.375 mm. The number of simulations for each stimulus level was 200: 40 random locations × 5 indentation depths. This adds up to 1600 simulated indentations for each of the two groups (seven levels of separation + 1 single pin × 40 random locations × 5 indentation depths).

To systematically assess the contribution of skin elasticity and afferent density, the simulations were run four times for: (i) the Young group; (ii) the Elderly group having lower YM but same afferent density as the Young group; (iii) the Elderly group having lower afferent density but same YM as the Young group; and (iv) the Elderly group with both lower afferent density and lower YM.

## Analysis of behavioural data and finger measurements

The aim was to determine whether skin biological elasticity, hydration and finger pad area are related to the performance obtained in the psychophysical task. First, the detection thresholds in the two groups were defined as the two-point distance at which the correct response rate was 75% (JNDs). To estimate this value, a Logistic function was fitted to individual data for each age group using MATLAB (Mathworks Inc., Natick, MA, USA) and Palamedes Toolbox, version 1.10.4 (https://www.palamedestoolbox.org). Second, the presence of a significant difference was assessed between the two age-groups in terms of finger properties, and the presence of correlation between each of the measured finger properties and the discrimination thresholds. Because of the presence of multicollinearity, commonality analysis (Mood, 1969, 1971; Nimon et al., 2008) was employed to gain a better understanding of the contribution of

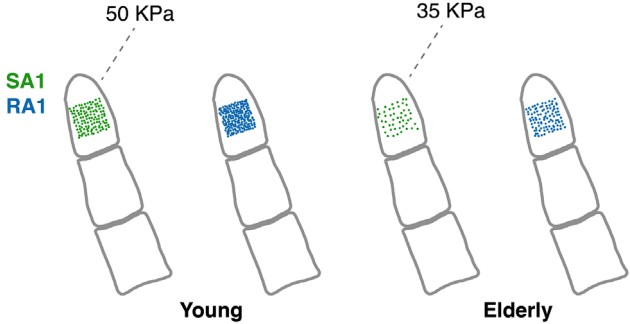

**Figure 2. Virtual SA1 and RA1 afferent units**
Virtual SA1 in green and RA1 in blue for Young (left) and Elderly (right) virtual groups modelled with TouchSim (Saal et al., 2017). The numbers at the top of the virtual finger show the modelled skin stiffness in KPa for each group. [Colour figure can be viewed at wileyonlinelibrary.com]

biological elasticity, hydration, finger pad area and age (or of all the possible combinations of these factors) to the variance observed in the two-point discrimination JNDs.

### Analysis of simulated neurophysiological data

A classification approach was used to discriminate between the simulated response to single pin and two pins at each separation level, separately. A linear discriminant analysis classifier was built with a 10-fold cross-validation measure using MATLAB built-in function *fitcdisc* with pseudolinear discrimination type to deal with predictors having zero within-class variance (e.g. first time-window when no spikes are elicited). Each classification was repeated 50 times to address the variability of the output because of the jittering of the stimulus location and the different indentation depths. The train-test split was randomised in each of the 50 iterations. The feature vector employed by the classifier consisted of the simulated spike count for each afferent. As a result of the high dimensionality of the feature space, we used principal component analysis to transform the original data and selected the first n-components that accounted for 95% of the explained variance. Principal component analysis was performed on the training set first and the obtained n-principal component coefficients (i.e. loadings) were used to transform the data in the test set. This procedure was repeated for each train-test split.

Based on the classifier output, the aim was to determine: (i) whether lower YM and lower afferent density negatively

affects the encoding of stimulus information; (ii) the accumulation of stimulus information throughout the indentation phase; and (iii) the contribution of the individual population of afferent fibres (SA1 and RA1) to the encoding of stimulus information, that is, which afferent type is more closely tuned to fine spatial details. To allow direct comparison with behavioural data, estimates of the stimulus level at which the classifier accuracy rate was 75% (referred to as JND) were obtained by fitting a logistic function to the accuracy values of the different classifications.

## Results

Alpha of 0.05 was set as criterion for statistical significance for all the reported tests.

### Psychophysics

The estimated JNDs revealed higher sensitivity for the Young group (mean ± SD: 0.69 ± 0.33 mm) compared to the Elderly group (mean ± SD: 2.49 ± 0.58 mm). For one young participant, it was not possible to estimate the psychometric curve parameters because the performance was above 75% even at 0.1 mm. The JND for this participant was set to 0 mm. A two-tailed independent sample $t$ test showed a significant difference between the two age-groups ($t_{26} = 10.081$; $P = 1.79 \times 10^{-10}$; Cohen's $d = 3.81$) (Fig. 3). The difference in the average JNDs

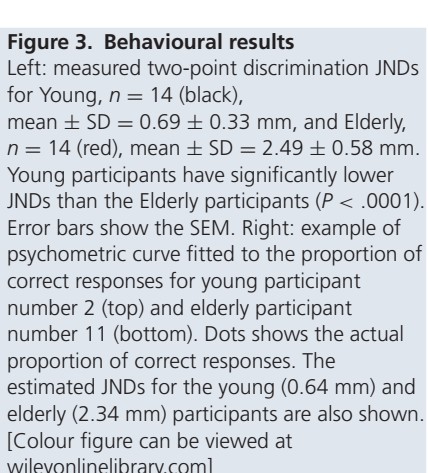

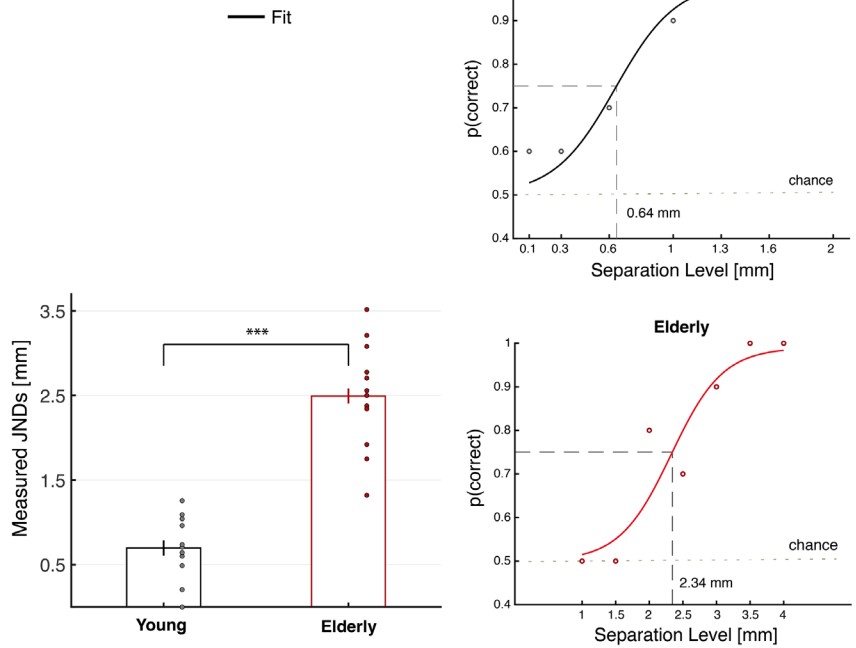

**Figure 3. Behavioural results**
Left: measured two-point discrimination JNDs for Young, $n = 14$ (black), mean ± SD = 0.69 ± 0.33 mm, and Elderly, $n = 14$ (red), mean ± SD = 2.49 ± 0.58 mm. Young participants have significantly lower JNDs than the Elderly participants ($P < .0001$). Error bars show the SEM. Right: example of psychometric curve fitted to the proportion of correct responses for young participant number 2 (top) and elderly participant number 11 (bottom). Dots shows the actual proportion of correct responses. The estimated JNDs for the young (0.64 mm) and elderly (2.34 mm) participants are also shown. [Colour figure can be viewed at wileyonlinelibrary.com]

**Table 1. Summary table of finger measurements**

|  | Young | Elderly |
| --- | --- | --- |
| Biological elasticity (Ur/Uf)** | 0.29 (0.10) | 0.19 (0.05) |
| Hydration (a.u.)*** | 76.9 (23.8) | 46.8 (17.6) |
| Finger pad area (mm²)** | 392.9 (70.3) | 473.0 (85.3) |

**$P < 0.01$, ***$P < 0.001$.

between the Elderly and the Young group was 1.79 mm [95% confidence interval (CI) = 1.43–2.16].

### Finger measurements

Mean ± SD values for biological elasticity, hydration and finger pad area are shown in Table 1. Three two-tailed independent sample $t$ tests showed that there was a statistically significant difference between Young and Elderly with respect to biological elasticity ($t_{26} = 2.985$; $P = 0.0061$; Cohen's $d = 1.13$; mean difference from Young to Elderly = 0.09; 95% CI = 0.03–0.15); a significant difference in hydration level ($t_{26} = 3.798$; $P = 0.00079$; Cohen's $d = 1.43$; mean difference from Young to Elderly = 30.08 a.u.; 95% CI = 13.80–46.37); and a significant difference in finger pad area ($t_{26} = 2.712$; $P = 0.0117$; Cohen's $d = 1.025$; mean difference from Elderly to Young = 80.1 mm²; 95% CI = 19.38–140.81).

A summary of the correlation analysis between each pair of finger properties and age is shown in Fig. 4. The results showed that all three variables were significantly correlated with age. Specifically, a negative relationship was found between hydration and age (Pearson's $r = -0.59$; $P = 0.00086$; 95% CI = −0.79 to −0.28), a negative relationship was found between

biological elasticity and age (Pearson's $r = -0.46$; $P = 0.0129$; 95% CI = −0.71 to −0.11) and a positive relationship was found between finger pad area and age (Pearson's $r = 0.53$; $P = 0.0038$; 95% CI = 0.19–0.75). In addition, biological elasticity was positively correlated with hydration (Pearson's $r = 0.58$; $P = 0.0012$; 95% CI = 0.26–0.78) and hydration was negatively correlated with finger pad area (Pearson's $r = -0.38$; $P = 0.0467$; 95% CI = −0.66 to −0.007). No significant correlation was found between biological elasticity and finger pad area (Pearson's $r = -0.06$; $P = 0.74$; 95% CI = −0.43 to 0.31).

### Influence of finger properties and age on behavioural task

The results of a correlation analysis performed between each of the measured finger properties and the estimated two-point discrimination JNDs showed a significant relationship for every pair (Fig. 5). In particular, there was a negative correlation between biological elasticity and the JNDs (Pearson's $r = -0.42$; $P = 0.0251$; 95% CI = −0.68 to −0.06), as well as between skin hydration and the JNDs (Pearson's $r = -0.61$; $P = 0.00059$; 95% CI = −0.80 to −0.30). A positive correlation was found between finger pad area and the JNDs (Pearson's $r = 0.56$; $P = 0.0019$; 95% CI = 0.24–0.77). A significant correlation was also found between age and the JNDs (Pearson's $r = 0.94$; $P = 6.45 \times 10^{-14}$; 95% CI = 0.87–0.97).

A commonality analysis performed on the various measures (Table 2) showed that biological elasticity, hydration, finger pad area and age explained 89.8% of the variance ($R^2$) in the behavioural performance measured through the two-point JND values. Examination of unique effects revealed that age was the best unique predictor of JNDs, accounting for 37.34% of the variance in the dependent variable. Finger pad area, hydration and biological elasticity explained 0.35%, 0.33% and 0.1% of the variance, respectively.

Although unique effects suggest that elasticity, hydration and finger pad area are not strongly related to JNDs, the analysis of common effects provide a more complete picture. In particular, the ratio between total effects for each predictor (i.e. unique and total of common effects combined) and the overall variance explained in

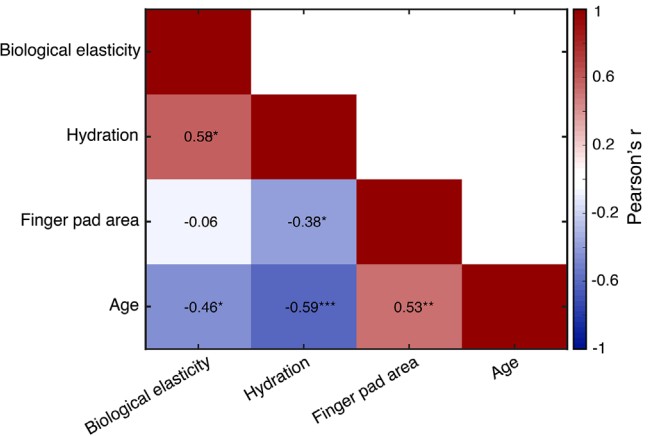

**Figure 4. Correlation analysis between finger properties and age**

Matrix showing the correlation results between each pair of the measured finger properties and between finger properties and age of participants. *$P < 0.05$, **$P < 0.01$, ***$P < 0.001$. [Colour figure can be viewed at wileyonlinelibrary.com]

**Table 2. Summary table of the commonality analysis results**

| | | | | Unique and common effects (%) | Total of common effects (%) |
|---|---|---|---|---|---|
| **Age** | | | | 37.34 | 51.55 |
| **Age** | **Hydration** | | | 14.4 | |
| **Age** | **Hydration** | **Area** | | 13.7 | |
| **Age** | | **Area** | **Elasticity** | 12.4 | |
| **Age** | | **Area** | | 5.6 | |
| **Age** | **Hydration** | **Area** | **Elasticity** | 4.9 | |
| **Age** | | | **Elasticity** | 2.6 | |
| | | **Area** | | 0.35 | 36.65 |
| | **Hydration** | | | 0.33 | 31.25 |
| | **Hydration** | **Area** | | 0.2 | |
| | **Hydration** | | **Elasticity** | 0.2 | |
| | | | **Elasticity** | 0.1 | 17.79 |
| | | **Area** | **Elasticity** | −0.1 | |
| | **Hydration** | **Area** | **Elasticity** | −0.1 | |
| **Age** | **Hydration** | | **Elasticity** | −2.1 | |
| *Total variance explained (%)* | | | | 89.8 | |

Values in the 'Unique and Common effects' column show the percentage of explained variance in the measured JNDs by each unique factor (unique effects, white) and by all possible combinations of factors (common effects, shades of grey). Total of common effects column shows the percentage of explained variance by all combinations of each factor with the others. Unique and common effects are listed in decreasing order

the behavioural performance was used to determine the amount of variance shared with regression effect by each independent variable. This calculation revealed that elasticity was involved with $(0.1 + 17.79)/89.8 = 19.9\%$ of the explained variance, finger pad area was involved with $(0.35 + 36.65)/89.8 = 41\%$, hydration was involved with $(0.33 + 31.25)/89.8 = 35\%$ and age contributed $(37.34 + 51.55)/89.8 = 98.9\%$.

These results showed that biological elasticity, hydration and finger pad area shared a significant amount of variance with the regression effect despite the major contribution of age.

**Figure 5. Correlation analysis between JNDs, finger properties and age**
Scatter plots showing the correlation between the two-point discrimination JNDs and each of the finger properties as well as age. Black dots represent values obtained from the participants in the Young group and red dots the ones from the Elderly group. Pearson's *r* and *P* values for each correlation are shown in the respective quadrant. [Colour figure can be viewed at wileyonlinelibrary.com]

## Simulated neurophysiological data

The spiking activity of SA1 and RA1 in response to the stimulus set used in the psychophysical task was simulated with three goals in mind: (1) determining whether and to what extent lower YM and lower afferent density affect the amount of information encoded in the afferents' response; (2) assessing how stimulus information accumulates over time for the Young group *vs.* the Elderly group; and (3) determining which afferent type is more tuned to fine spatial details, again comparing Young and Elderly group. An example of the simulated spiking response of SA1 and RA1 fibres is shown in Fig. 8*B*. As expected, the response pattern differs between the two types of fibres. SA1 fibres showed a strong activity at the onset of the stimuli and some sustained response during the hold phase, whereas RA1 fibres were active only during the onset and offset of the indentation.

Figure 6 shows the population spike count for each simulated group at each stimulus level, including the single pin, at a fixed indentation depth of 2.5 mm and collapsed across the randomised contact points. Increasing separation level between the two-pins produced an overall increase in spike count for all groups. However, the variability of the spike count across stimuli depended on the simulated group. For the Young group, the number of spikes generated in response to the single pin is similar to the one obtained for the two-pins separated by 0.1 and 0.3 mm (i.e. black line and shaded area in Fig. 6), whereas it differs substantially from the spike count in response to the two-pins separated by

1 mm or more. Instead, it can be seen that, for the Elderly group with both manipulations, this is not the case and there is a great overlapping in terms of population spike count between all stimulus levels.

In brief, the highest variability was observed for the Young group, followed by the Elderly group with lower YM, the Elderly group with lower afferent density, and last the Elderly group with both manipulations.

With this in mind, the simulated response over the entire 500 ms stimulation window was used to generate the classification output for the simulated Young group and each of the three manipulations for the Elderly group to test (1). This analysis was performed by using the response of SA1 and RA1 together to discriminate between the single pin and two pins at each separation level, separately. Then, a logistic curve was fitted to the classifier accuracy results to estimate the simulated two-point discrimination JNDs (75% correct response rate) for each virtual group. The mean ± SD estimated JNDs for the Young group was $0.87 \pm 0.04$ mm, the mean ± SD JNDs for the Elderly group having lower YM and same afferent density as the Young group was $0.99 \pm 0.04$ mm, the mean ± SD JNDs for the Elderly group having lower afferent density and same YM as the Young group was $1.25 \pm 0.04$ mm and, finally, the mean ± SD JNDs for the Elderly group with both lower YM and lower afferent density was $1.35 \pm 0.07$ mm (Fig. 7*A*). A one-way ANOVA was performed to compare the effect of our manipulations on the virtual JNDs.

The results showed that there was a statistically significant difference between at least two of the simulated JNDs ($F_{3,196} = 1045.63$; $P = 2.76 \times 10^{-120}$; $\eta^2 = 0.94$). Tukey's honestly significant difference test for multiple comparison revealed that the virtual JNDs were significantly smaller for the Young group than each of the three Elderly groups ($P = 3.77 \times 10^{-9}$ for all comparisons). The estimated mean difference in JND between the Young group and the Elderly group with lower YM was 0.12 mm (95% CI = 0.15–0.10; Cohen's $d$ = 3.11). The estimated mean difference in JND between the Young group and the Elderly group with lower afferent density was 0.38 mm (95% CI = 0.41–0.36; Cohen's $d$ = 9.61). The estimated mean difference between the Young group and the Elderly group with both lower YM and lower afferent density was 0.48 mm (95% CI = 0.51–0.45; Cohen's $d$ = 8.39). In addition, the JNDs for the Elderly group with both manipulations was significantly higher than the JNDs for the Elderly group with lower afferent density ($P = 3.77 \times 10^{-9}$; estimated mean difference of 0.10 mm; 95% CI = 0.12–0.07) and the JNDs for the Elderly group with lower YM ($P = 3.77 \times 10^{-9}$; estimated mean difference 0.36 mm; 95% CI = 0.38–0.33). Finally, the JNDs for the Elderly group with lower afferent density were significantly higher than the JNDs for the Elderly group with lower YM ($P = 3.77 \times 10^{-9}$; estimated mean

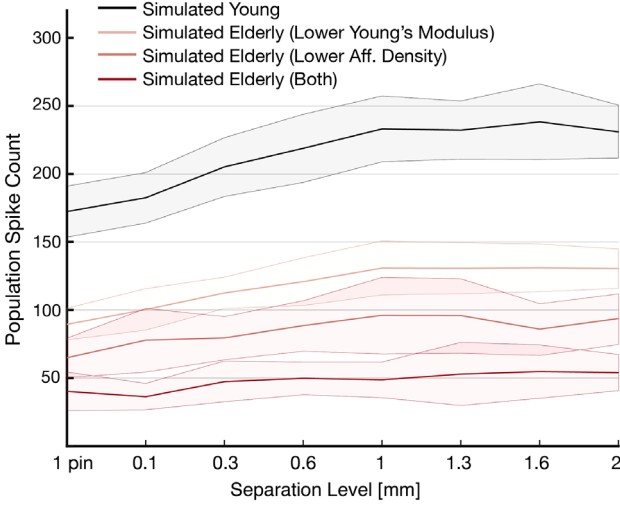

**Figure 6. Simulated population spike count**
Population spike count for all the simulated groups at each stimulus level for 2.5 mm indentation depth collapsed across the 40 randomised contact points. The shaded area shows the SD. The variability within the same stimulus level is mainly driven by the jittering of stimulus location. [Colour figure can be viewed at wileyonlinelibrary.com]

difference 0.26 mm; 95% CI = 0.28–0.23). These results showed an additive effect of each of our manipulations with a stronger contribution of afferent density.

To establish how the stimulus information accumulates throughout the indentation period (2), classification analysis was performed over 15 logarithmically spaced cumulative time windows, ranging from 1 ms to 500 ms. The response of SA1 and RA1 together was used to discriminate between the single pin and two pins at each separation level, separately. This analysis was run on each separation level for the Young group and the Elderly group. Similar to the previous analysis, the virtual JNDs (75% correct response rate) were estimated based on classification results by fitting a Logistic curve to the accuracy values of the decoder. Figure 8A shows the evolution of the virtual JNDs for young and elderly over time.

For the Young group, the spatial activation across afferents was informative enough to discriminate between the single pin and the two pins as early as 20 ms after the onset of indentation, when only a few spikes have been elicited. The results for the Elderly group showed not only a poorer classification performance, as previously shown, but also a slower information build-up with the neural response only becoming informative at around 35 ms (Fig. 8A, red line). A further difference between simulated young and elderly data was the variability of results, indicated by the standard deviation (Fig. 8A, shaded area). In particular, we observed that results from the young were more consistent than the elderly counterpart.

Finally, we set out to determine which afferent type was more tuned to the fine spatial layout of the virtual stimuli. The simulated response over the entire 500 ms stimulation window was used to generate the classification

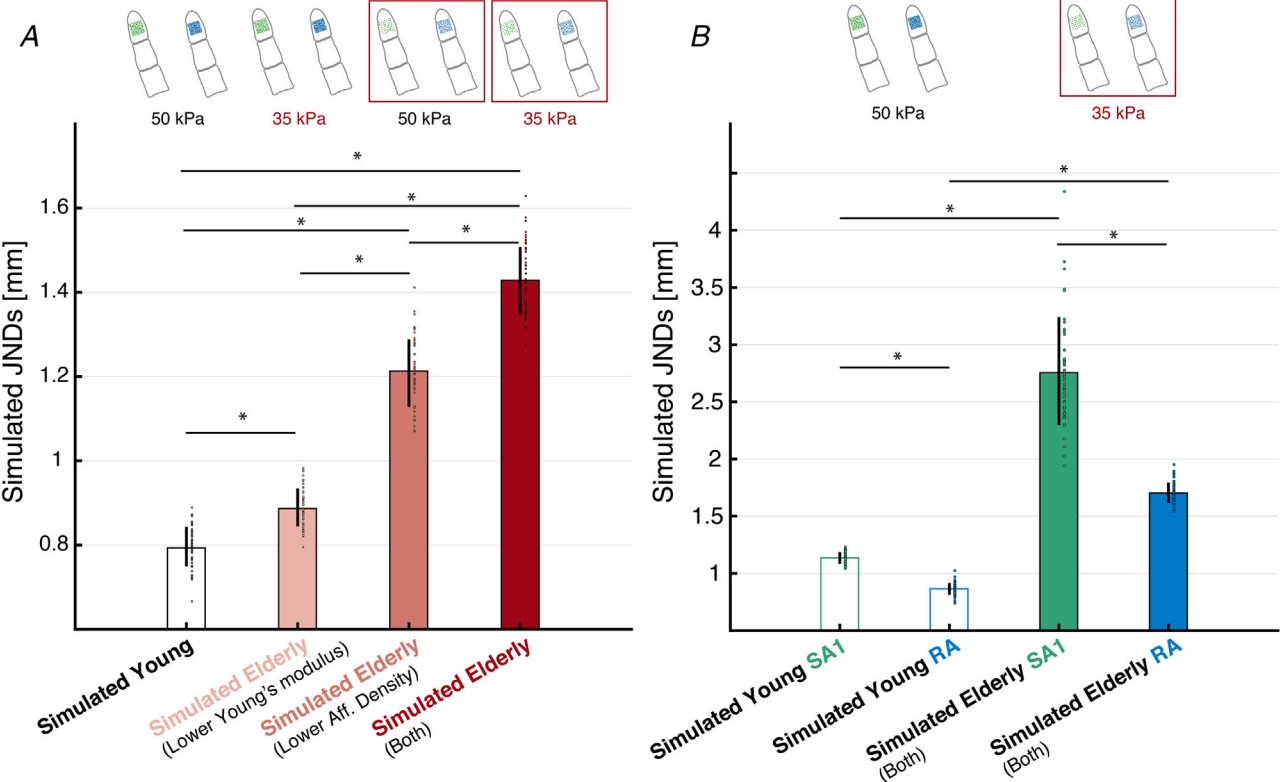

**Figure 7. JNDs estimated from simulations output**
Simulated JNDs based on accuracy values obtained from classification of the simulated response to the two-point discrimination task using parameters to capture the Young and Elderly groups. Classifier built on the entire indentation window (i.e. 500 ms). *A*, JNDs estimated from the response of SA1 and RA combined for Young (black), Elderly with lower Young's modulus (pink), Elderly with lower afferent density (light red), and Elderly with both lower Young's modulus and lower afferent density (red). Error bars represent the SD of the JNDs obtained for each of the 50 runs of classification. Coloured dots show the values obtained for each of the 50 runs of classification. *P < 0.0001, Tukey's honestly significant difference test. *B*, JNDs estimated from the response of SA1 (green) and RA1 (blue), separately. Void bars show results for the virtual Young group. Filled bars for the virtual Elderly group with both lower Young's modulus and lower afferent density. Coloured dots represent the estimates from each of the 50 classification runs. Error bars represent the SD of the JNDs obtained for each of the 50 runs of classification. *P < 0.0001, Bonferroni corrected.
[Colour figure can be viewed at wileyonlinelibrary.com]

output for the simulated Young and Elderly group, built on SA1 data and RA1 data separately. Then, the JNDs (75% correct response rate) were estimated for both group and each afferent type (Fig. 7*B*). A two-way mixed ANOVA was performed to evaluate the effect of age and receptor type on the estimated virtual JNDs. Receptor type was treated as within-subject factor and age was treated as between-subject factor. The results showed a significant main effect of receptor type ($F_{1,98} = 771.585$; $P = 2.98 \times 10^{-48}$; $\eta^2 = 0.887$) and a significant main effect of age ($F_{1,98} = 1870.828$; $P = 1.17 \times 10^{-65}$; $\eta^2 = 0.95$). Additionally, there was a significant interaction between receptor type and age ($F_{1,98} = 122.389$; $P = 6.07 \times 10^{-19}$; $\eta^2 = 0.555$).

Two dependent sample *t* tests were performed to compare the simulated JNDs based on SA1 against RA1 data in each age group, and two independent sample *t* tests were used to compare the simulated JNDs based on the SA1 and RA1 response of the Young group against the respective JNDs obtained for the Elderly group. *P* values were corrected with the Bonferroni method. The results showed that, in both age groups, there was a significant difference between the JNDs estimated from the response of SA1 and those estimated from the response of RA1. For the Young group (SA1-RA; $t_{49} = 43.7661$; $P_{\text{corrected}} = 2.44 \times 10^{-40}$; Cohen's $d = 8.91$; estimated mean difference $= 0.39$ mm; 95% CI $= 0.37$–$0.41$). For the Elderly group (SA1-RA; $t_{49} = 19.7842$; $P_{\text{corrected}} = 2.07 \times 10^{-24}$; Cohen's $d = 3.88$; estimated mean difference $= 0.91$ mm; 95% CI $= 0.82$–$1.00$). As expected, a significant difference was found between the JNDs estimated on SA1 response of Young and Elderly group (SA1$_{\text{young}}$ – SA1$_{\text{elderly}}$; $t_{98} = -28.03$; $P_{\text{corrected}} = 5.41 \times 10^{-48}$; Cohen's $d = 5.61$; estimated mean difference $= -1.31$ mm; 95% CI $= -1.40$ to $-1.22$) and between the JNDs estimated on RA response of Young and Elderly group (RA$_{\text{young}}$ – RA$_{\text{elderly}}$; $t_{98} = -80.77$; $P_{\text{corrected}} = 7.09 \times 10^{-91}$; Cohen's $d = 16.15$; estimated mean difference $= -0.79$ mm; 95% CI $= -0.81$ to $-0.77$).

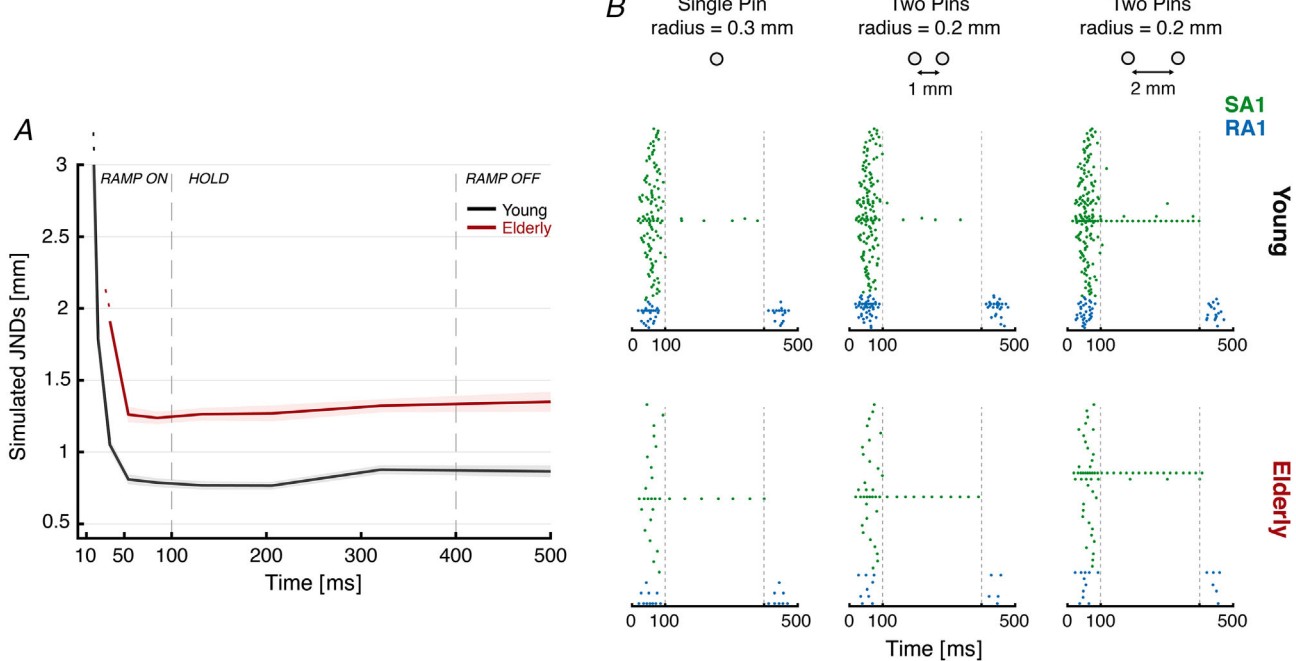

**Figure 8. Time-course estimation of JNDs based on simulations output and raster plots of the simulated neural activity**
*A*, time-course estimation of virtual JNDs based on simulated neurophysiological data for the Young group (black lines, SA1 and RA1 units combined) and for the Elderly group (red line, SA1 and RA1 units combined). The shaded area represents the SD of the estimated JNDs based on each of the 50 classification runs. JND values at 500 ms correspond to the performance shown in Fig. 7*A* for the Young group (black) and the Elderly group with both manipulations (dark red). *B*, simulated population response of SA1 (green) and RA1 (blue) tactile fibres for Young (top row) and Elderly group (bottom row) to a subset of virtual stimuli indented 2.5 mm. From left to right: raster plots show the neural response to a single pin with a radius of 0.3 mm, two pins with a radius of 0.2 mm each separated by 1 mm and two pins with a radius of 0.2 mm each separated by 2 mm. Only the active fibers (at least one spike) were plotted. Vertical grey dotted lines indicate the start and the end of the hold phase of the trace. [Colour figure can be viewed at wileyonlinelibrary.com]

## Discussion

A combination of psychophysics with measurements of finger properties and computational modelling was employed to evaluate some of the potential factors underlying the deterioration of tactile spatial sensitivity observed with ageing. The results from the previous literature on two-point threshold showing a decrease in performance in the elderly population for statically indented tactile stimuli (Kalisch et al., 2009; Stevens, 1992; Woodward, 1993) were replicated by using passive stimulation with computer-controlled speed, force and contact duration. The Elderly group two-point discrimination threshold was over three times larger compared to the Young group. Importantly, new evidence was provided on which factors might drive the observed difference in the psychophysical task between the two groups. Commonality analysis showed that age was the best unique predictor of the sensitivity to our set of stimuli, followed by hydration, finger pad area and biological elasticity. Although the unique contribution of skin properties to the regression effect was minimal, biological elasticity, hydration and finger pad area definitely contributed to the regression effect as they were involved with 19.9%, 35% and 41% of the total explained variance in the measured two-point JNDs, respectively.

The relevance of age in predicting the behavioural performance is not surprising because ageing is accompanied not only by a decline of peripheral sensory systems, but also an alteration of cortical representations of sensory information, as well as cognitive processes. Previous studies showed that impaired tactile acuity in the elderly is reflected in increased excitability of the primary somatosensory area S1 (Lenz et al., 2012) and enlargement of hand representation (Kalisch et al., 2009). This might be because of a reduction of intracortical inhibition that results in the spread of activity from the stimulated cortical RFs to nearby regions (Kalisch et al., 2009) and may elicit less sharp neural representations. Memory decay and sustained attention might also have played a role in this kind of paradigm. Our 2IFC task involved short-term memory to store the sensory information about the first stimulus and compare it to the second one a few seconds later. Similarly, the repetitive nature of task could make attending to the sensory inputs more challenging for the older participants. The cognitive demand might have been exacerbated in the elderly by the poorer sensory inputs. Ageing effects on central mechanisms might also explain the difference observed between the behavioural thresholds in which more than a three-fold increase of threshold for the Elderly group was recorded and the simulated thresholds where a two-fold increase was found.

The contribution of finger pad area to the perceptual performance can be linked to the fact that the number of at least one type of mechanoreceptor, Meissner's corpuscles, does not vary much across individuals and is inversely correlated with finger pad area (Dillon et al., 2001; Nolano et al., 2003). The Elderly group had a larger finger pad area than young participants. One possible reason for this difference might be the enlargement of bone epiphysis observed in both males and females with age (Kalichman et al., 2008), This in itself could contribute to reducing density of receptors with age. However, ageing is also characterised by a reduction in the number and also changes in the morphology and depth of receptors (García-Piqueras et al., 2019), as well as slower conduction velocities of peripheral nerves because of demyelination (Peters, 2002), which might also degrade tactile input and raise two-point discrimination thresholds with age. Importantly, the loss of Meissner's corpuscles varies across elderly individuals and probably contributes to the extent to which tactile sensitivity is reduced with age (Skedung et al., 2018)

The role of finger pad area and age was consistent with the results from the simulation based on Saal et al. (2017). Model-generated JNDs were negatively influenced mainly by lower afferent density, which relates to either factor. Indeed, the density of the virtual afferents had a greater impact on the simulation results than the manipulation of YM. This is not surprising according to the traditional view that tactile spatial resolution is limited by the afferent density and the centre-to-centre spacing of the receptive fields (Dodson et al., 1998; Friedman et al., 2002). However, the extent to which these factors can affect spatial acuity is still not clear.

SA1 and RA1 fibres are characterised by innervation branching, which generates receptive fields with a complex sensitivity map with multiple hotspots (Johansson, 1978). In the model of Saal et al. (2017), the receptive field of these neurons are circular with a single sensitive zone. However, they have similar features to the actual ones, including overall size, sensitivity to indentation depth and lower threshold in the centre of the receptive field compared to the periphery. Some insights on the contribution of the complex structure of receptive fields to a different measure of spatial acuity come from the work of Hay and Pruszynski (2020). They compared a model of RA1 units having such a complex receptive field with a similar model of RA1 having uniform receptive fields. They found that the multiple hotspots slightly improved the discrimination of fine orientations (e.g. $-1°$ *vs.* $+1°$) compared to a similar model of receptive fields with uniform sensitivity. However, the classification was largely above chance for both uniform and complex receptive fields. In addition, the modelling work of Hay and Pruszynski (2020) does not include any consideration of skin mechanical response and the stresses acting on the receptors.

Regarding skin elasticity and hydration, the results suggested that they also contributed to the discrimination

performance. Simulation results also showed a small effect of YM thatr is related to biological elasticity. In particular, lower YM significantly increased the estimated JNDs by only 0.12 mm compared to the virtual Young group. The significant results in the presence of a small effect could have been driven by the relatively high number of observations generated with simulations that were needed for the classification of the stimuli and the low variability of the model output. In addition, this minimal difference might not be meaningful at behavioural level (e.g. discriminating or manipulating objects).

It is worth noting that attempts to measure YM of skin have provided contrasting results. In a recent review paper, Kalra and Lowe (2016) showed that the estimated YM depends on the methods used to measure it as well as other factors. Indentation and torsion tests appear to result in estimates of YM that reduce with age (Boyer et al., 2009), whereas suction tests sometimes provide increasing estimates with age (Diridollou et al., 2001). Because this experiment involved indentation of the stimuli, a lower value for the Elderly group was chosen from the literature. Although skin elasticity was measured with a suction device, the obtained measurements across age groups were similar to those of Boyer et al. (2009) who found thatr lower elasticity, as measured in the elderly with a suction device, was associated with lower stiffness (i.e. Young's modulus), as measured with an indentation protocol. This choice was also supported by the work of Yang et al. (2018) who used ultrasound to measure YM and found that it was significantly lower in the over 50 year olds compared to the 20−50-year-old group. Yang et al. (2018) also observed that the age related difference in YM was greater at the finger pad than at other body sites.

The analysis of the simulated neurophysiological data over time provided evidence for effects of YM and afferent density not only on the asymptotic level of performance, but also on how the stimulus information unfolds over time. Classification output showed that stimulus information is available as early as 20 ms from the initial contact for the Young group. This finding is consistent with Delhaye et al. (2019) who showed a similar unfolding of information in an edge orientation task. As noted, having such a rapid response is crucial for object manipulation and fine manual dexterity tasks, although the precise temporal dynamics are dependent on the indentation parameters and can occur at slightly different temporal scales. Importantly, comparing these results with those obtained for the Elderly group revealed that our manipulations delayed the way that stimulus information builds up. This might interact and further reduce elderly performance compared to young, if, for example, the contact with the stimuli is very brief.

Regarding the individual contribution of SA1 and RA1 units in conveying the shape of the statically indented stimuli, there is still an open debate about whether the functions of different unit types are segregated (Johnson, 2001; Johnson et al., 2000) or partially overlapping (Saal & Bensmaia, 2014). The results for both the Young and the Elderly virtual groups suggest that the spatial activation and firing rates of both SA1 and RA1 fibres carry information about the stimulus spatial layout. These findings support the rate coding hypothesis by which geometric features are encoded in the firing rate intensity and variations across afferents. It has been shown that the firing rate of SA1 and to a less extent RA1 for both static and dynamic touch might convey the shape of a step (i.e. steepness) indented either vertically or stroked across the skin (Srinivasan & LaMotte, 1987), the curvature of corrugated surfaces (LaMotte & Srinivasan, 1996), the orientation of cylinders (Dodson et al., 1998) and the configuration of raised dots (Connor & Johnson, 1992). Overall, the results support the idea that these types of fibre are of an equal importance, which is in contrast to the traditional segregation model (Johnson, 2001; Johnson et al., 2000). The latter assigns an exclusive role for shape perception to the SA1. However, recent studies are in line with our findings supporting a convergence of functions across the different afferent types (Weber et al., 2013). To address this question, it will be necessary to study the population activity of real tactile neurons as this may reveal emergent properties that are not present in the response of individual units.

The model of Saal et al. (2017) has been built on several assumptions to simplify the computations to generate the neural response. The skin is modelled as flat, homogenous and elastic, with isotropic behaviour, and does not include any hard structures (e.g. bone), nor fingerprints. Having a realistic 3D shape and layered structure of the finger is definitely important for large deformations when the applied force and contact area would cause the skin to protrude from both sides of the finger pad activating receptors in that area. The stimuli in this experiment were flat-ended pins with small diameter (0.2 and 0.3 mm) indented to small depths (1–2.5 mm) in the centre of the virtual finger and the assumed flat structure of the skin in the model does well in reproducing the response properties of the simulated afferents to this type of stimuli as shown in Saal et al., 2017. In particular, the firing rate and the spike timing correlate well with actual data showing several response properties of real first-order neurons including slow *vs.* fast adaptation, frequency tuning and edge enhancement/surround suppression.

The lack of fingerprint geometry, anisotropic behaviour and viscoelastic response would be a major concern for dynamic stimuli in the presence of friction (e.g. sliding movement). For example, the skin deforms to a different extent if a scanning movement occurs in lateral *vs.* proximo-distal direction or when making contact with a sticky or slippery texture. Indeed, in this model, the virtual stimuli are defined as a single cylindrical pin or a

set of pins that can be indented only orthogonally, which makes it suitable for simulating the classical two-point discrimination task in which the skin is stimulated with static stimuli in a vertical direction without the presence of any major shear component. Nonetheless, the contribution of the realistic 3D shape of the finger and the hard structures (i.e. bones, nails) to the informativeness of the neural activation cannot be assessed with this model.

PC units were not included in the simulations because their large receptive fields and sparse distribution makes them unsuited for resolving the fine spatial layout of tactile stimuli. This is highlighted in the seminal work of Phillips et al. (1988) who showed that the PC afferent response used to construct a spatial event plot result in a blurred image. Nonetheless, PC sensitivity to vibratory stimuli across a wide range of frequency and very small amplitude skin deformation is important for the perception of textures and vibrations acting on objects held in the hand (Kandel et al., 2021), as well as the contact onset and offset of static stimuli.

In the present study, simulations results were based on the spatial activation of the afferent populations, or the population firing rate (i.e. rate coding). This does not imply that this information can be directly used by the central nervous system. Other potential coding strategies have been proposed as mechanism to extract shape information. These include the spike timings of individual afferents for edge orientation (Pruszynski & Johansson, 2014) and the variations across afferents of the first spike latency for curvature (Johansson & Birznieks, 2004). However, the temporal aspect of the neural response has been shown to be highly susceptible to differences in other stimulus parameters (Suresh et al., 2016) and taking into account precise spike timings to classify statically indented edges with different orientation might make the signal less informative if these differences are present (Delhaye et al., 2019) compared to the population firing rates. Certainly, spike timing is essential in the coding of texture and vibrations (Mackevicius et al., 2012; Weber et al., 2013) and the present findings do not exclude the possibility that the stimulus spatial layout can also be extracted from the timing of individual spikes. In addition, the neural response evolves as it moves through the different stages of the hierarchy (i.e. spinal cord, brainstem, thalamus, cortex). An open question is where the integration and transformation of the signals coming from the four types of afferent units begins (for a review see Abraira and Ginty, 2013). For example, the integration of the signal from a population of neurons at the level of the cuneate nucleus is compatible with a coincidence detection mechanism based on the timing of individual spikes (Pruszynski & Johansson, 2014). Future work should aim at elucidating the contribution of precise spike timings with a focus on ageing touch, including skin properties, the response properties of aging neurons and the integration of the afferent signal starting at the level of the cuneate nucleus.

## Conclusions

Spatial tactile sensitivity decreases throughout the lifespan. Although the role of peripheral sensory components, such as skin and mechanoreceptors properties, has been highlighted, there is little evidence that link these factors to the deterioration of tactile perception. This experiment confirmed that elderly people have lower two-point discrimination sensitivity than their younger counterparts. Importantly, this difference was linked to finger pad area, which was found to be higher, implying lower afferent density, in the Elderly group. There were also contributions of reducing biological elasticity and hydration to ageing reductions in tactile sensitivity, although these were appreciably less than for the finger pad area. The present findings also highlight the contribution of a general ageing effect that might include impaired cognitive processes and it is suggested that this may have contributed to the difference between behavioural and simulation results.

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

## Additional information

### Data availability statement

The data that support the findings of this study are available from the corresponding author upon reasonable request.

### Competing interests

The authors declare that they have no competing interests.

### Author contributions

D.D., M.D.L. and A.M.W. developed the research question. D.D. designed and performed the experiments and analysed the data. D.D., A.M.W. and M.D.L. wrote the paper.

### Funding

This work was funded by a collaborative studentship between the University of Birmingham and Procter and Gamble and a BBSRC grant BB/R003971/1 to A. M. Wing.

### Acknowledgements

We thank Claudio Zito for developing the Delta robot framework, Hannes Saal and Benoit Delhaye for discussions about the TouchSim model; Winnie Chua for statistical advice; Marta Arroyo for her valuable comments on an earlier version of the paper; and Aldrin Loomes for the line drawing of the experimental setup.

### Keywords

ageing, cutaneous afferent, modelling, simulations, skin, tactile perception

## Supporting information

Additional supporting information can be found online in the Supporting Information section at the end of the HTML view of the article. Supporting information files available:

**Statistical Summary Document**
**Peer Review History**

