## [Peer Review History · The Journal of Physiology]

Skin properties and afferent density in the deterioration of tactile spatial acuity with age

Davide Deflorio, Massimiliano Di Luca, and Alan Miles Wing
DOI: 10.1113/JP283174

Corresponding author(s): Alan Wing (a.m.wing@bham.ac.uk)

The following individual(s) involved in review of this submission have agreed to reveal their identity: Hakan Olausson (Referee #2)

Review Timeline:

Submission Date:	21-Feb-2022
Editorial Decision:	23-Mar-2022
Resubmission Received:	27-Oct-2022
Editorial Decision:	15-Nov-2022
Revision Received:	01-Dec-2022
Accepted:	13-Dec-2022

Senior Editor: Richard Carson

Reviewing Editor: Vaughan Macefield

Transaction Report:

Dear Dr. Wing,

Re: JP-RP-2022-282927 "Skin stiffness and afferent density in the deterioration of tactile spatial acuity with age" by Davide Deflorio, Massimiliano Di Luca, and Alan Wing

Thank you for submitting your manuscript to The Journal of Physiology. It has been assessed by a Reviewing Editor and by 2 Referees and the reports are copied below.

Please let your co-authors know of the following editorial decision as quickly as possible.

As you will see, in its current form, the manuscript is not acceptable for publication in The Journal of Physiology. In comments to me, the Reviewing Editor expressed interest in the potential of this study, but much work still needs to be done (and this may include new experiments) in order to satisfactorily address the concerns raised in the reports.

In view of this interest, I would like to offer you the opportunity to carry out all of the changes requested in full, and to resubmit a new manuscript using the "Submit Special Case Resubmission for JP-RP-2022-282927..." on your homepage.

We cannot, of course, guarantee ultimate acceptance at this stage as the revisions required are substantial. However, we encourage you to consider the requested changes and resubmit your work to us if you are able to complete or address all changes.

A new manuscript would be renumbered and redated, but the original referees would be consulted wherever possible. An additional referee's opinion could be sought, if the Reviewing Editor felt it necessary. A full response to each of the reports should be uploaded with a new version.

I hope that the points raised in the reports will be helpful to you.

Yours sincerely,

Richard Carson
Senior Editor
The Journal of Physiology

EDITOR COMMENTS

Reviewing Editor:

Thank you for submitting your manuscript to The Journal of Physiology. I have now received reports from two independent reviewers, both experts in the field of human tactile neurophysiology and psychophysics. As you will see, Reviewer 1 has some recommendations for improvement but Reviewer 2 has major concerns with the modelling aspects of the study and hence the validity of the conclusions. In particular, this reviewer is concerned that there is low agreement between the simulations and the human psychophysical experiments, as well as the suitability of the model used in the simulations the paper. As such, I'm afraid the manuscript is not suitable as is, and will require significant rewriting in order to bring it up to the standard required of a modelling paper, particularly given that it does not include an assessment of the actual firing properties of human tactile afferents in young and aged participants. It would also be wise to include the data from Srinivasan and LaMotte in the monkey and Condon et al in the human on the effects of compliance on force encoding by tactile afferents.

REFEREE COMMENTS

Referee #1:

The authors present a highly-controlled and well-designed study investigating skin and tactile perception changes on the glabrous hand skin with age. The manuscript is interesting and very well-written, as well as displaying the data well (e.g. showing individual participant data in the graphs). I especially very much like their approach to 2-point discrimination testing, which at times can lead to problems of interpretation via the stimulation and the paradigm. Here, this is an excellent setup for many reasons, which minimizes errors and gives clear interpretations of the data. I have a few comments below that may help improve the paper more.

In the abstract, they say that the 'density of mechanoreceptors decreases', whereas I think it would be better to say 'mechanoreceptive afferents', as it is very likely that there is axon degradation as well, not just at the end of the receptor. Also, on p.4 of the introduction, the authors rightly cover the loss of mechanoreceptors in the skin, but the whole afferent can degrade.

I think the authors need to be more specific right at the beginning of the introduction about what skin they are talking about. Skin is highly heterogeneous and it is likely that aging affects it differently, for example, see the classic Stevens and Choo work (1996, Somatosensory & Motor Res), which not only covers tactile tests on the glabrous hand skin, but all over the body, and how this changes over the lifespan. I think it would be good to add this study to the manuscript, but also start by saying that you are focusing on the glabrous hand skin.

At the beginning of the methods, please include details about whether written informed consent was gained and whether the study received approval from an ethical committee.

At the start of the 'Psychophysical task and stimulation setup' of the methods, please add more details about the exact location tested on the right index finger. Was this exactly the same between participants? The outer fingertip tends to be drier with age and there are subtle differences in sensitivity over the fingertip, due to mechanoreceptor density differences (e.g. Johansson & Vallbo, 1979, J Physiol), thus it is particularly important here to give this information. It could be more pertinent in aging, where the addition of RA2 signals during touch could aid the interpretation of the other tactile signals.

To follow up on this in the discussion, I think it is relevant to discuss the potential future inclusion of RA2 and SA2 afferents. The model of Saal et al. does not include SA2s (as it is based on monkey work, where they are lacking, but these are readily found in humans), thus this information is not available. I can also understand the authors' reasoning (e.g. methods where they say PC/RA2 afferents are 'not tuned to fine spatial details'); however, these type 2 afferents would certainly help underpin touch perception and they should not be discounted. I agree that the focus here on type 1 afferents is good, but it would be worthwhile discussing the addition of type 2 afferents and their possible contribution for future work.

Why was finger pad area significantly larger in the Elderly group? Can you comment more on this in the discussion? Does the finger change shape with age?

Please spell out RA1 and SA1 when first used in the text.

Referee #2:

The current paper is trying to elucidate the reasons for deteriorating spatial acuity that comes with age. To do this they have performed psychophysical experiments and modeling of skin dynamics. Conceptually this is important research that would contribute to the missing understanding of how skin dynamics affect the tactile sense. The psychophysical method and description are good, and the use of the robotic stimulator is a nice way to ensure adequate reproducibility. However, the paper is severely lacking in clarity both with respect to the neurophysiological reasoning and use of statistical methods and how these are presented. Another limitation is that the model does not account for how the response properties of single mechanoreceptors change with age in humans.

Major issues

The simulated responses to the tactile stimuli were made using the model by Saal et al. The model is only referred to and is stated to be "extended". If this extension is only the manipulation of Young's modulus or there are more to it is never made clear. Furthermore, there is no description of the model apart from the reference which assumes that the reader is fully aware of the particulars of the model used. However, if one is familiar with the Saal-model then there are several aspects that is discussed within the model paper which is of relevance to this paper. For example, in the Saal paper they explicitly state that they only include "the stress perpendicular to the skin surface" while in the current paper the longitudinal stress is discussed. The chosen model should be discussed in the current paper since a large portion of the paper is dependent on the model. The already mentioned potential issue with the perpendicular stress for example, is this a problem?

Furthermore, in the human experiments stimuli presentation is CONTACT ON-SLIDE-CONTACT OFF while the simulations have CONTACT ON-MOVE-CONTACT OFF. This is, probably, due to the fact that the Saal method cannot handle sliding stimulation; an aspect that is not even mentioned in the current paper. This is of course an important difference since the sliding event is a fundamental haptic dimension and could easily be argued to make a big difference (see "Is there a 'plenhaptic' function?" by Hayward for an in-depth discussion, <https://doi.org/10.1098/rstb.2011.0150>). How does this affect the outcome of the paper?

The analysis of the simulated data is presented in a very shallow manner. The spike metric Victor-Purpura distance is presented and the q cost is set to zero, because the temporal structure of a spike response "does not provide relevant additional information". Which is a strange argument to present since the stimulation during the human experiments is dynamic and the stimulation during the simulations are also dynamic (to a low degree) since it is RAMP-HOLD-RAMP. Furthermore, there are numerous counter arguments to the notion of rate coding and these arguments shows that tactile primary afferent processing (that for the upper trunk in humans ends up in the cuneate nucleus) is highly dependent on the temporal structure (see <https://doi.org/10.1016/j.neuron.2014.07.038> and <https://doi.org/10.3389/fncel.2018.00210>). Please discuss this more in detail and elucidate in which way the current paper avoids being dependent on the function of the Cuneate nucleus. Furthermore, there is no discussion in relation to this why VP-distance was chosen of all the available spike metrics (for a good overview see <https://dx.doi.org/10.3389%2Ffncom.2019.00082>). Finally, it is not known how the sensor responses change with respect to age and is definitively not included in the Saal-model. What is the argument with respect to this that makes the use of the model relevant?

After the introduction of the VP-metric follows a very rough description of the classification process where all the responses of the sensors are thrown into RSS. Then converted into Euclidean coordinates, which begs the question what that is? The common way to describe it is to state that the space is Euclidean, not the coordinates.

Furthermore, it is stated that MDS is used to "convert the data" into the Euclidean coordinates (space), which begs the question what it was before? Since classical MDS assumes that the input is Euclidean.

Then the concept of "principal components" is written without explanation, followed by the statement that the PCs that accounted for 95% of the variance was selected. How this selection process was constructed is never written.

Based on these PCs the classifier is then built. If this is true, then the authors have built in auto-correlation into their analysis since they have extracted the PCs from ALL data, and then classify the data based in that. They state that they do the test-train split AFTER the PCs has been created. This means that the PCs will include the training data. It is also not written if there is any difference between the following 50 repetitions of the LDA classification (I assume that the test-train split is randomized each time, but this is never stated and is also moot if the PCs have been defined prior to the split). Finally, LDA is never explained. All in all, the classification algorithm presented is unclear and potentially auto-correlated which in that case would mean that all the results are nonsense. Is the above written understanding correct? If that is the case, please explain how the analysis does not exhibit auto-correlation.

When the authors analyze how the stimulus information accumulates they do so for multiple durations for the Young group, but only one for the Elderly group without a rationale behind this. They also find that based on their simulations their classification performance is above chance already at 0.1 mm level which is not in line with their experimental data. This might be a strong argument why there is no apparent translational power between their experimental data and their simulated data. Furthermore, they use the word "spatiotemporal" in the same section, which is misleading since they use

VP-distance with a q cost of zero meaning that the temporal aspect is close to zero. This also indicate that the pure number of spikes is what they measure. So, they for transparence should show the evolution of the number of spikes for the different stimulus paradigms. Why only one duration for the Elderly group? How is the analyzed data spatiotemporal?

In conclusion, the paper would need a major rewrite with respect to the complexities inherent in skin biomechanics and the related sensor responses. Furthermore, all the descriptions of the analysis methods used, and the consequent results would need a major rewrite and in some, potential cases, a complete change. Finally, there is no apparent agreement between the simulation and the psychophysical results which begs the question what the simulation brings to the table?

Minor

No discussion of ethics approval is present. Even if there was concluded that no Ethics approval was needed, the rationale behind this should be included.

There is no spread metric in the introduction where mean ages are presented. If a mean is written, then a spread needs to be written as well.

Throughout the paper Confidence Intervals are missing, making the understanding of the p-values very incomplete.

The authors say that they perform seven two-tailed t-tests. They accurately correct the significance level but they 1) fail to report the actual p-values, 2) fail to report the Confidence intervals, and 3) do not use the corrected p-values in figure 9.

ADDITIONAL FORMATTING REQUIREMENTS:

-You must start the Methods section with a paragraph headed Ethical Approval. If experiments were conducted on humans confirmation that informed consent was obtained, preferably in writing, that the studies conformed to the standards set by the latest revision of the Declaration of Helsinki, and that the procedures were approved by a properly constituted ethics committee, which should be named, must be included in the article file. If the research study was registered (clause 35 of the Declaration of Helsinki) the registration database should be indicated, otherwise the lack of registration should be noted as an exception (e.g. The study conformed to the standards set by the Declaration of Helsinki, except for registration in a database.). For further information see: <https://physoc.onlinelibrary.wiley.com/hub/human-experiments>

-Please upload separate high-quality figure files via the submission form.

-A Statistical Summary Document, summarising the statistics presented in the manuscript, is required upon revision. It must be on the Journal's template, which can be downloaded from the link in the Statistical Summary Document section here: https://jp.msubmit.net/cgi-bin/main.plex?form_type=display_requirements#statistics

-Papers must comply with the Statistics Policy https://jp.msubmit.net/cgi-bin/main.plex?form_type=display_requirements#statistics

In summary:

-If n {less than or equal to} 30, all data points must be plotted in the figure in a way that reveals their range and distribution. A bar graph with data points overlaid, a box and whisker plot or a violin plot (preferably with data points included) are acceptable formats.

-If $n > 30$, then the entire raw dataset must be made available either as supporting information, or hosted on a not-for-profit repository e.g. FigShare, with access details provided in the manuscript.

-'n' clearly defined (e.g. x cells from y slices in z animals) in the Methods. Authors should be mindful of pseudoreplication.

-All relevant 'n' values must be clearly stated in the main text, figures and tables, and the Statistical Summary Document (required upon revision)

-The most appropriate summary statistic (e.g. mean or median and standard deviation) must be used. Standard Error of the Mean (SEM) alone is not permitted.

-Exact p values must be stated. Authors must not use 'greater than' or 'less than'. Exact p values must be stated to three significant figures even when 'no statistical significance' is claimed.

-Statistics Summary Document completed appropriately upon revision

-A Data Availability Statement is required for all papers reporting original data. This must be in the Additional Information section of the manuscript itself. It must have the paragraph heading "Data Availability Statement". All data supporting the results in the paper must be either: in the paper itself; uploaded as Supporting Information for Online Publication; or archived in an appropriate public repository. The statement needs to describe the availability or the absence of shared data. Authors must include in their Statement: a link to the repository they have used, or a statement that it is available as Supporting Information; reference the data in the appropriate section(s) of their manuscript; and cite the data they have shared in the References section. Whenever possible the scripts and other artefacts used to generate the analyses presented in the paper should also be publicly archived. If sharing data compromises ethical standards or legal requirements then authors are not expected to share it, but must note this in their Statement. For more information, see our Statistics Policy.

-Please include an Abstract Figure. The Abstract Figure is a piece of artwork designed to give readers an immediate understanding of the research and should summarise the main conclusions. If possible, the image should be easily 'readable' from left to right or top to bottom. It should show the physiological relevance of the manuscript so readers can assess the importance and content of its findings. Abstract Figures should not merely recapitulate other figures in the manuscript. Please try to keep the diagram as simple as possible and without superfluous information that may distract from the main conclusion(s). Abstract Figures must be provided by authors no later than the revised manuscript stage and should be uploaded as a separate file during online submission labelled as File Type 'Abstract Figure'. Please ensure that you include the figure legend in the main article file. All Abstract Figures should be created using BioRender. Authors should use The Journal's premium BioRender account to export high-resolution images.

Follow this link <https://app.biorender.com/portal/jphysiol> and enter your details and ensure you enter the manuscript number JP-RP-2022-282927 to be directed to enter our premium site. Select a figure type when creating the figure so The Journal of Physiology logo appears. When you have completed your figure(s) download and then upload as the figure file(s) for your revised submission. If you choose not to take up this offer we require figures to be of similar quality and resolution. If you are opting out of this service to authors, state this in the Comments section on the Detailed Information page of the submission form.

EDITOR COMMENTS

Reviewing Editor:

I have now received reports from two independent reviewers, both experts in the field of human tactile neurophysiology and psychophysics. As you will see, Reviewer 1 has some recommendations for improvement but Reviewer 2 has major concerns with the modelling aspects of the study and hence the validity of the conclusions. In particular, this reviewer is concerned that there is low agreement between the simulations and the human psychophysical experiments, as well as the suitability of the model used in the simulations the paper. As such, I'm afraid the manuscript is not suitable as is, and will require significant rewriting in order to bring it up to the standard required of a modelling paper, particularly given that it does not include an assessment of the actual firing properties of human tactile afferents in young and aged participants. It would also be wise to include the data from Srinivasan and LaMotte in the monkey and Condon et al in the human on the effects of compliance on force encoding by tactile afferents.

- We would like to thank the editor and the reviewers for their work. We have performed the changes suggested by the reviewers, including accounting for the concerns of reviewer 2 which affected the validity of the conclusion.

The major changes are: (1) use of commonality analysis to understand the contribution to the discrimination judgments of the various factors (2) simplification of the analysis of simulated neurophysiological data (3) extension of the simulation to include evidence accumulation in elderly group.

REFEREE COMMENTS

Referee #1:

The authors present a highly-controlled and well-designed study investigating skin and tactile perception changes on the glabrous hand skin with age. The manuscript is interesting and very well-written, as well as displaying the data well (e.g. showing individual participant data in the graphs). I especially very much like their approach to 2-point discrimination testing, which at times can lead to problems of interpretation via the stimulation and the paradigm. Here, this is an excellent setup for many reasons, which minimizes errors and gives clear interpretations of the data. I have a few comments below that may help improve the paper more.

In the abstract, they say that the 'density of mechanoreceptors decreases', whereas I think it would be better to say 'mechanoreceptive afferents', as it is very likely that there is axon degradation as well, not just at the end of the receptor. Also, on p.4 of the introduction, the authors rightly cover the loss of mechanoreceptors in the skin, but the whole afferent can degrade.

- We have now changed the term to mechanoreceptive afferents and accounted for the degradation of the axon with the appropriate literature in the introduction.

I think the authors need to be more specific right at the beginning of the introduction about what skin they are talking about. Skin is highly heterogenous and it is likely that aging affects it differently, for example, see the classic Stevens and Choo work (1996, Somatosen & Motor Res), which not only covers tactile tests on the glabrous hand skin, but all over the body, and how this changes over the lifespan. I think it would be good to add this study to the manuscript, but also start by saying that you are focusing on the glabrous hand skin.

- We have made clear that we are talking about the glabrous skin of the finger pad and added the suggested paper (Stevens and Choo, 1996).

At the beginning of the methods, please include details about whether written informed consent was gained and whether the study received approval from an ethical committee.

- We have added this information.

At the start of the 'Psychophysical task and stimulation setup' of the methods, please add more details about the exact location tested on the right index finger. Was this exactly the same between participants? The outer fingertip tends to be drier with age and there are subtle differences in sensitivity over the fingertip, due to mechanoreceptor density differences (e.g. Johansson & Vallbo, 1979, J Physiol), thus it is particularly important here to give this information. It could be more pertinent in aging, where the addition of RA2 signals during touch could aid the interpretation of the other tactile signals.

- We have specified the location in the Methods section 'Psychophysical task and stimulation setup'.

To follow up on this in the discussion, I think it is relevant to discuss the potential future inclusion of RA2 and SA2 afferents. The model of Saal et al. does not include SA2s (as it is based on monkey work, where they are lacking, but these are readily found in humans), thus this information is not available. I can also understand the authors' reasoning (e.g. methods where they say PC/RA2 afferents are 'not tuned to fine spatial details'); however, these type 2 afferents would certainly help underpin touch perception and they should not be discounted. I agree that the focus here on type 1 afferents is good, but it would be worthwhile discussing the addition of type 2 afferents and their possible contribution for future work.

- We have now discussed the role of PC afferents in the Discussion section.

Why was finger pad area significantly larger in the Elderly group? Can you comment more on this in the discussion? Does the finger change shape with age?

- We have commented on this in the Discussion.

Please spell out RA1 and SA1 when first used in the text.

- We have introduced the abbreviations.

Referee #2:

The current paper is trying to elucidate the reasons for deteriorating spatial acuity that comes with age. To do this they have performed psychophysical experiments and modeling of skin dynamics. Conceptually this is important research that would contribute to the missing understanding of how skin dynamics effect the tactile sense. The psychophysical method and description are good, and the use of the robotic stimulator is a nice way to ensure adequate reproducibility. However, the paper is severely lacking in clarity both with respect to the neurophysiological reasoning and use of statistical methods and how these are presented. Another limitation is that the model does not account for how the response properties of single mechanoafferents change with age in humans.

- We think there has been a misinterpretation of our experiment design, as there isn't a shear component in the stimulation used in this study. We have attempted to clarify this in the manuscript.

- We also provided a better explanation of the neurophysiological reasoning behind the simulations and more details regarding analysing the simulated neurophysiological data.

Major issues

The simulated responses to the tactile stimuli were made using the model by Saal et al. The model is only referred to and is stated to be "extended". If this extension is only the manipulation of Youngs modulus or there are more to it is never made clear. Furthermore, there is no description of the model apart from the reference which assumes that the reader is fully aware of the particulars of the model used. However, if one is familiar with the Saal-model then there are several aspects that is discussed within the model paper which is of relevance to this paper.

- We now introduce Saal's model in a new section titled 'Overview of TouchSim' model, which also includes the details related to the work on this paper. We have clarified how the model has been modified.

For example, in the Saal paper they explicitly state that they only include "the stress perpendicular to the skin surface" while in the current paper the longitudinal stress is discussed. The chosen model should be discussed in the current paper since a large portion of the paper is dependent on the model. The already mentioned potential issue with the perpendicular stress for example, is this a problem?

- We have now clarified that the Saal's model and the conditions in our experiment involve only stimuli applied perpendicular to the skin. We took out the term 'longitudinal' that was used inappropriately to avoid confusion. We also argued that this model is suitable for this application as no major longitudinal stress is produced by our static stimuli.

Furthermore, in the human experiments stimuli presentation is CONTACT ON-SLIDE-CONTACT OFF while the simulations have CONTACT ON-MOVE-CONTACT OFF. This is, probably, due to the fact that the Saal method cannot handle sliding stimulation; an aspect that is not even mentioned in the current paper. This is of course an important difference since the sliding event is a fundamental haptic dimension and could easily be argued to make a big difference (see "Is there a 'plenhaptic' function?" by Hayward for an in-depth discussion, <https://doi.org/10.1098/rstb.2011.0150>). How does this affect the outcome of the paper?

-We share the concerns of the reviewer and we have added this limitation to the paragraph about Saal model. However, this was not initially mentioned as it does not apply to our experiment.

The analysis of the simulated data is presented in a very shallow manner.

-Thank you for pointing this out. We agree there's need for more detail and we take this opportunity to make the analysis of simulation data clearer and reproducible.

The spike metric Victor-Purpura distance is presented and the q cost is set to zero, because the temporal structure of a spike response "does not provide relevant additional information". Which is a strange argument to present since the stimulation during the human experiments is dynamic and the stimulation during the simulations are also dynamic (to a low degree) since it is RAMP-HOLD-RAMP.

-We have provided a better explanation to why the focus was on the rate coding and why the temporal structure of neural response was not assessed.

Furthermore, there are numerous counter arguments to the notion of rate coding and these arguments shows that tactile primary afferental processing (that for the upper trunc in humans ends up in the cuneate nucleus) is highly dependent on the temporal structure (see <https://doi.org/10.1016/j.neuron.2014.07.038> and <https://doi.org/10.3389/fncel.2018.00210>) Please discuss this more in detail and elucidate in which way the current paper avoids being dependent on the function of the Cuneate nucleus.

-We have included in the discussion limitations related to looking at first-order neurons alone without taking into account second-order neurons where signal processing is likely to begin. We have also discussed why a rate code was chosen for investigation rather than a temporal code (i.e., spike timing)

Furthermore, there is no discussion in relation to this why VP-distance was chosen of all the available spike metrics (for a good overview see <https://dx.doi.org/10.3389%2Ffncom.2019.00082>).

-We have now simplified the analysis that no longer needs any spike metrics. This simplifies the description and reproducibility of our approach.

Finally, it is not known how the sensor responses change with respect to age and is definitively not included in the Saal-model. What is the argument with respect to this that makes the use of the

model relevant?

-We have provided a rationale behind the modelling part of the paper in the introduction and discussed its implications in the discussion section.

After the introduction of the VP-metric follows a very rough description of the classification process where all the responses of the sensors are thrown into RSS. Then converted into Euclidean coordinates, which begs the question what that is? The common way to describe it is to state that the space is Euclidean, not the coordinates.

Furthermore, it is stated that MDS is used to "convert the data" into the Euclidean coordinates (space), which begs the question what it was before? Since classical MDS assumes that the input is Euclidean.

Then the concept of "principal components" is written without explanation, followed by the statement that the PCs that accounted for 95% of the variance was selected. How this selection process was constructed is never written.

Based on these PCs the classifier is then built. If this is true, then the authors have built in auto-correlation into their analysis since they have extracted the PCs from ALL data, and then classify the data based in that. They state that they do the test-train split AFTER the PCs has been created. This means that the PCs will include the training data.

It is also not written if there is any difference between the following 50 repetitions of the LDA classification (I assume that the test-train split is randomized each time, but this is never stated and is also moot if the PCs have been defined prior to the split). Finally, LDA is never explained. All in all, the classification algorithm presented is unclear and potentially auto-correlated which in that case would mean that all the results are nonsense. Is the above written understanding correct? If that is the case, please explain how the analysis does not exhibit auto-correlation.

When the authors analyze how the stimulus information accumulates they do so for multiple durations for the Young group, but only one for the Elderly group without a rationale behind this.

- We believe the misunderstanding stems from the use of the term Principal Component to refer to the coordinates obtained from the MDS. We are sorry we used a term that is unclear in this context. In brief, we did not perform PCA for dimensionality reduction which, the reviewer is correct, would have been an issue if pca was performed on all data before the train/test split. We have taken this opportunity to change the approach for the analysis of the neurophysiological data by using PCA instead of the long pipeline used in the first place. This part was rewritten ensuring more clarity in regard to the details of the analysis.

They also find that based on their simulations their classification performance is above chance already at 0.1 mm level which is not in line with their experimental data. This might be a strong argument why there is no apparent translational power between their experimental data and their simulated data.

- Thanks for pointing this out. We have tried to clarify why we believe there is translational power between experimental and simulated data.

Our focus was to estimate the separation level at which the correct response rate is 75%. Participants performance was estimated as the stimulus level (i.e. separation distance) at which they could respond correctly 75% of the time. This was done by fitting a Logistic curve to the

response correct rate at each separation level. Similarly, we estimated the separation level at which the classifier performed correctly 75% of the time. Results show a good match between the behavioural and simulated JNDs . We have now reported all the analysis in terms of JNDs. Importantly, the classification performance slightly above chance at 0.1 mm is not in contrast with the behavioural data where some of our participants could discriminate the 2 pins from the single pin even at 0.1 mm.

Furthermore, they use the word "spatiotemporal" in the same section, which is misleading since they use VP-distance with a q cost of zero meaning that the temporal aspect is close to zero. This also indicate that the pure number of spikes is what they measure. So, they for transparency should show the evolution of the number of spikes for the different stimulus paradigms. Why only one duration for the Elderly group? How is the analyzed data spatiotemporal?

-We changed it to spatial and clarified that is the number of spikes we are looking at. We modified the analysis to allow a better comparison between young and elderly.

In conclusion, the paper would need a major rewrite with respect to the complexities inherent in skin biomechanics and the related sensor responses. Furthermore, all the descriptions of the analysis methods used, and the consequent results would need a major rewrite and in some, potential cases, a complete change. Finally, there is no apparent agreement between the simulation and the psychophysical results which begs the question what the simulation brings to the table?

Minor

No discussion of ethics approval is present. Even if there was concluded that no Ethics approval was needed, the rationale behind this should be included.

- We have added this information

There is no spread metric in the introduction where mean ages are presented. If a mean is written, then a spread needs to be written as well.

- We have added spread metric for the study by Skedung et al. (2018). However, spread metric was not reported in the remaining papers cited in the introduction.

Throughout the paper Confidence Intervals are missing, making the understanding of the p-values very incomplete.

The authors say that they perform seven two-tailed t-tests. They accurately correct the significance level but they 1) fail to report the actual p-values, 2) fail to report the Confidence intervals, and 3) do not use the corrected p-values in figure 9.

- We have now added Confidence interval for all mean comparisons and reported corrected p-values.

Dear Dr. Wing,

Re: JP-RP-2022-283174X "Skin properties and afferent density in the deterioration of tactile spatial acuity with age" by Davide Deflorio, Massimiliano Di Luca, and Alan Wing

Thank you for submitting your manuscript to The Journal of Physiology. It has been assessed by a Reviewing Editor and by 2 expert referees and we are pleased to tell you that it is acceptable for publication following minor revision.

REVISION CHECKLIST:

- 'Potential Cover Art' for consideration as the issue's cover image

- Appropriate Supporting Information (Video, audio or data set: see https://jp.msubmit.net/cgi-bin/main.plex?form_type=display_requirements#supp).

We look forward to receiving your revised submission.

Yours sincerely,

Richard Carson
Senior Editor
The Journal of Physiology

REQUIRED ITEMS:

- You must start the Methods section with a paragraph headed Ethical Approval. If experiments were conducted on humans confirmation that informed consent was obtained, preferably in writing, that the studies conformed to the standards set by the latest revision of the Declaration of Helsinki, and that the procedures were approved by a properly constituted ethics committee, which should be named, must be included in the article file. If the research study was registered (clause 35 of the Declaration of Helsinki) the registration database should be indicated, otherwise the lack of registration should be noted as an exception (e.g. The study conformed to the standards set by the Declaration of Helsinki, except for registration in a database.). For further information see: <https://physoc.onlinelibrary.wiley.com/hub/human-experiments>.

- Papers must comply with the Statistics Policy: https://jp.msubmit.net/cgi-bin/main.plex?form_type=display_requirements#statistics.

In summary:

- If $n \leq 30$, all data points must be plotted in the figure in a way that reveals their range and distribution. A bar graph with data points overlaid, a box and whisker plot or a violin plot (preferably with data points included) are acceptable formats.

- If $n > 30$, then the entire raw dataset must be made available either as supporting information, or hosted on a not-for-profit repository e.g. FigShare, with access details provided in the manuscript.

- 'n' clearly defined (e.g. x cells from y slices in z animals) in the Methods. Authors should be mindful of pseudoreplication.

- All relevant 'n' values must be clearly stated in the main text, figures and tables, and the Statistical Summary Document (required upon revision).

- The most appropriate summary statistic (e.g. mean or median and standard deviation) must be used. Standard Error of the Mean (SEM) alone is not permitted.

- Exact p values must be stated. Authors must not use 'greater than' or 'less than'. Exact p values must be stated to three significant figures even when 'no statistical significance' is claimed.

- Statistics Summary Document completed appropriately upon revision.

- Please include an Abstract Figure file, as well as the figure legend text within the main article file. The Abstract Figure is a piece of artwork designed to give readers an immediate understanding of the research and should summarise the main conclusions. If possible, the image should be easily 'readable' from left to right or top to bottom. It should show the physiological relevance of the manuscript so readers can assess the importance and content of its findings. Abstract Figures should not merely recapitulate other figures in the manuscript. Please try to keep the diagram as simple as possible and without superfluous information that may distract from the main conclusion(s). Abstract Figures must be provided by authors no later than the revised manuscript stage and should be uploaded as a separate file during online submission labelled as File Type 'Abstract Figure'. Please ensure that you include the figure legend in the main article file. All Abstract Figures should be created using BioRender. Authors should use The Journal's premium BioRender account to export high-resolution images. Details on how to use and access the premium account are included as part of this email.

EDITOR COMMENTS

Reviewing Editor:

Thank you for submitting your manuscript to The Journal of Physiology. I have now received reports from two independent reviewers, both experts in human somatosensory neurophysiology. As you will see, both reviewers thought highly of your manuscript, noting that deterioration of tactile function with age is understudied and that your manuscript contributes with an in-depth mechanistic analysis. However, both reviewers raise some issues that you will need to address. I look forward to receiving your revised manuscript in due course.

REFeree COMMENTS

Referee #1:

I thank the authors for providing clear responses in the review process and making the necessary changes. I have a few further comments.

In the abstract, 'density of mechanoreceptive afferent decreases' should be 'density of mechanoreceptive afferents decreases' (with the 's' at the end of afferent).

Was the present study pre-registered? If not, in the ethics part of the methods, please add, 'apart from registration in a database', as per the Declaration of Helsinki (#35 <https://www.wma.net/policies-post/wma-declaration-of-helsinki-ethical-principles-for-medical-research-involving-human-subjects/>).

In the paragraph on Meissner corpuscles in the discussion, the authors could add the paper by Skedung et al (2018, Sci Rep), as they show that some elderly participants had conserved numbers of Meissner's (and were better at tactile tests), but the majority had decreased numbers. It is likely that these factors are also variable with age.

Referee #2:

Several of the concerns raised in the first review has been mitigated. A major change to the analysis has been performed and is overall an improvement. It is easier to follow the method and less concerns are raised.

However, a few concerns are present in the updated manuscript. Below are major and minor issues.

Major issues:

Page 27-28: Commonality analysis:

I have several concerns that is, at least unclear to me although they might be due to misunderstandings rather than errors.

The numbers in table 2 and the numbers in the paragraph starting with "Although unique effects suggest that elasticity," are not the same. Every numerator differs from the table. Elasticity is 17.89 in the paragraph, 17.79 in the table; finger pad area is 37 in the paragraph, 36.65 in the table; hydration is 31.58 in the paragraph and 31.25 in the table; age is 88.9 in the paragraph and 51.55 in the table. After some reading and calculating I assume that "unique and total of common effects combined" means that the authors added the Coefficient column to the Total of Common Effects column, which produces the numbers used in the text. If this is the case, then there should be a correspondence of terms used. Thus, the relationship between table and paragraph is unclear.

Furthermore, it is never explained why this metric is relevant. From the numbers in the table, it looks like the main variable associated with the JND variance is Age. Looking at the plots in fig.5 this makes sense, but on the other hand the magnitude of the impact of the remaining variables seems unreasonably low. Are they so colinear with age that they do not matter or is it perhaps an issue with improper normalization prior to the commonality analysis?

Also, it is unclear what the relationship is between the initial claim that age accounted for 37% of the variance, and the last claim of the following paragraph where it is said to account for 98.9% of the variance. Which is it?

This is repeated in the first paragraph of the Discussion, any alteration in the preceding text must be reflected there as well.

Overall, I think that the explanatory paragraph on page 28 is unclear, and it is very hard to find the correspondence between the numbers in the text and the numbers in the referred table.

Figure 6 right: The magnitude of the SD is heavily influenced by the magnitude of the mean value. I.e., the potential for larger variations increases with increasing frequency. Converting the SDs in fig. 6. Right plot into Z-score (SD/Mean) seems to convert the variability into the 15-20% range for all the groups (calculations made from visual inspection of the plots). The last statement of the first paragraph on page 30 can be questioned. Since the classification is made based on the rate-code (i.e., number of spikes) then the variability will mostly introduce noise since it is quite obvious from the left panel in fig.6 that

there is a roughly 2 SD difference between the Means of each of the Elderly groups, and even more up to the Young group. Please clarify how the variability improves information content with respect to rate-coding and why SD is preferred over z-score?

Fig. 8 discusses the stimulus information build up over time. As it is rate-code it means how the number of spikes accumulates. We know from Fig.6 that the mean number of spikes from the different stimulus contexts produce quite different number of spikes. It would be interesting to see the typical responses from each of the groups. Like a raster plot or similar. This would make it more accessible and relevant for peers that work with recorded data from primary afferents.

Minor issues:

Page 25: Multiple comparisons without preceding test or p-value correction. This in contrast to the later figure 7 and related texts where suddenly both bonferroni correction and pre-test are used.

Varying use of exact and approximate p-values.

No definition of what "significant" means.

Page 26 paragraph starting with "A summary of...": Missing CI for "No significant correlation was found between biological elasticity and finger pad area (Pearson's $r = -0.06$, $p = .74$)."

Mixing of styles: 360% increase and two-fold in increase, page 35.

END OF COMMENTS

1st Confidential Review

27-Oct-2022

EDITOR COMMENTS

Reviewing Editor:

Thank you for submitting your manuscript to The Journal of Physiology. I have now received reports from two independent reviewers, both experts in human somatosensory neurophysiology. As you will see, both reviewers thought highly of your manuscript, noting that deterioration of tactile function with age is understudied and that your manuscript contributes with an in-depth mechanistic analysis. However, both reviewers raise some issues that you will need to address. I look forward to receiving your revised manuscript in due course.

REFEREE COMMENTS

Referee #1:

I thank the authors for providing clear responses in the review process and making the necessary changes. I have a few further comments.

In the abstract, 'density of mechanoreceptive afferent decreases' should be 'density of mechanoreceptive afferents decreases' (with the 's' at the end of afferent).

- **We corrected this.**

Was the present study pre-registered? If not, in the ethics part of the methods, please add, 'apart from registration in a database', as per the Declaration of Helsinki (#35 <https://www.wma.net/policies-post/wma-declaration-of-helsinki-ethical-principles-for-medical-research-involving-human-subjects/>).

- **We added this information.**

In the paragraph on Meissner corpuscles in the discussion, the authors could add

the paper by Skedung et al (2018, Sci Rep), as they show that some elderly participants had conserved numbers of Meissner's (and were better at tactile tests), but the majority had decreased numbers. It is likely that these factors are also variable with age.

- We have added the suggested paper in the Discussion section.

Referee #2:

Several of the concerns raised in the first review has been mitigated. A major change to the analysis has been performed and is overall an improvement. It is easier to follow the method and less concerns are raised.

However, a few concerns are present in the updated manuscript. Below are major and minor issues.

Major issues:

Page 27-28: Commonality analysis:

I have several concerns that is, at least unclear to me although they might be due to misunderstandings rather than errors.

The numbers in table 2 and the numbers in the paragraph starting with "Although unique effects suggest that elasticity," are not the same. Every numerator differs from the table. Elasticity is 17.89 in the paragraph, 17.79 in the table; finger pad area is 37 in the paragraph, 36.65 in the table; hydration is 31.58 in the paragraph and 31.25 in the table; age is 88.9 in the paragraph and 51.55 in the table. After some reading and calculating I assume that "unique and total of common effects combined" means that the authors added the Coefficient column to the Total of Common Effects column, which produces the numbers used in the text. If this is the case, then there should be a correspondence of terms used. Thus, the relationship between table and paragraph is unclear.

- **We have changed the term used in the table to match the one used in the text.**

Furthermore, it is never explained why this metric is relevant. From the numbers in the table, it looks like the main variable associated with the JND variance is Age. Looking at the plots in fig.5 this makes sense, but on the other hand the magnitude of the impact of the remaining variables seems unreasonably low. Are they so colinear with age that they do not matter or is it perhaps an issue with improper normalization prior to the commonality analysis?

Also, it is unclear what the relationship is between the initial claim that age accounted for 37% of the variance, and the last claim of the following paragraph where it is said to account for 98.9% of the variance. Which is it?

This is repeated in the first paragraph of the Discussion, any alteration in the preceding text must be reflected there as well.

Overall, I think that the explanatory paragraph on page 28 is unclear, and it is very hard to find the correspondence between the numbers in the text and the numbers in the referred table.

- **We thank the reviewer for pointing this out and we tried to clarify the text in the manuscript to avoid confusion between R^2 , which is the regression effect (percentage of variance in the dependent variable that can be explained by our model), the percentage of explained variance in the dependent variable by each unique predictor, and the percentage of variance shared by each predictor with the regression effect (i.e., the ratio between the percentage of explained variance in the dependent variable by each unique predictor plus all combinations of each predictor with the others and the overall variance explained in the behavioural performance).**

- **The low impact of the remaining variables is due to the collinearity issue which is the reason why we used commonality analysis to provide a more accurate picture of the relationship between predictors and dependent variable. Although the remaining variable provided little unique contribution to the regression effect, they shared a significant amount of variance with the regression effect as shown by the ratio between total effects for each predictor plus all combinations of each predictors with the others and the overall variance explained in the behavioural performance.**

Figure 6 right: The magnitude of the SD is heavily influenced by the magnitude of the mean value. I.e., the potential for larger variations increases with increasing frequency. Converting the SDs in fig. 6. Right plot into Z-score (SD/Mean) seems to convert the variability into the 15-20% range for all the groups (calculations made from visual inspection of the plots). The last statement of the first paragraph on page 30 can be questioned. Since the classification is made based on the rate-code (i.e., number of spikes) then the variability will mostly introduce noise since it is quite obvious from the left panel in fig.6 that there is a roughly 2 SD difference between the Means of each of the Elderly groups, and even more up to the Young group. Please clarify how the variability improves information content with respect to rate-coding and why SD is preferred over z-score?

- **We thank the reviewer for pointing this out. We have now removed the figure and text related to the SD as we realised it was superfluous. We have highlighted that the population spike count is more variable across stimuli for the simulated Young group than the other three groups without the need for any additional measure (e.g. SD). Also, this is not exactly what the classifier is using. The information comes from the variability of the firing rate across individual afferents and not the overall (population) spike count.**

Fig. 8 discusses the stimulus information build up over time. As it is rate-code it means how the number of spikes accumulates. We know from Fig.6 that the mean number of spikes from the different stimulus contexts produce quite different number of spikes. It would be interesting to see the typical responses from each of the

groups. Like a raster plot or similar. This would make it more accessible and relevant for peers that work with recorded data from primary afferents.

- **We have added raster plots in Figure 8 showing an example of the simulated neural response in the two-age groups.**

Minor issues:

Page 25: Multiple comparisons without preceding test or p-value correction. This in contrast to the later figure 7 and related texts where suddenly both bonferroni correction and pre-test are used.

- **We have included a two-way mixed ANOVA before the multiple comparison.**

Varying use of exact and approximate p-values.

- **We have now reported exact p-values in all tests.**

No definition of what "significant" means.

- **We have now specified: "Alpha of 0.05 was set as criterion for statistical significance" at the beginning of the Results section.**

Page 26 paragraph starting with "A summary of...": Missing CI for "No significant correlation was found between biological elasticity and finger pad area (Pearson's $r = -0.06$, $p = .74$)."

- **We have added this information.**

Mixing of styles: 360% increase and two-fold in increase, page 35.

- **We have now used the same style.**

Dear Dr Wing,

Re: JP-RP-2022-283174XR1 "Skin properties and afferent density in the deterioration of tactile spatial acuity with age" by Davide Deflorio, Massimiliano Di Luca, and Alan Miles Wing

We are pleased to tell you that your paper has been accepted for publication in The Journal of Physiology.

Authors should note that it is too late at this point to offer corrections prior to proofing. The accepted version will be published online, ahead of the copy edited and typeset version being made available. Major corrections at proof stage, such as changes to figures, will be referred to the Editors for approval before they can be incorporated. Only minor changes, such as to style and consistency, should be made at proof stage. Changes that need to be made after proof stage will usually require a formal correction notice.

Yours sincerely,

Richard Carson
Senior Editor
The Journal of Physiology

P.S. - You can help your research get the attention it deserves! Check out Wiley's free Promotion Guide for best-practice recommendations for promoting your work at www.wileyauthors.com/eeo/guide. You can learn more about Wiley Editing Services which offers professional video, design, and writing services to create shareable video abstracts, infographics, conference posters, lay summaries, and research news stories for your research at www.wileyauthors.com/eeo/promotion.

IMPORTANT NOTICE ABOUT OPEN ACCESS: To assist authors whose funding agencies mandate public access to published research findings sooner than 12 months after publication, The Journal of Physiology allows authors to pay an Open Access (OA) fee to have their papers made freely available immediately on publication.

You can check if your funder or institution has a Wiley Open Access Account here: <https://authorservices.wiley.com/author-resources/Journal-Authors/licensing-and-open-access/open-access/author-compliance-tool.html>.

EDITOR COMMENTS

Reviewing Editor:

Thank you for attending to the remaining concerns of Reviewer 2, who is now satisfied with your amendments.

REFeree COMMENTS

Referee #1:

The authors have answered all my comments well, I have no further comments, and I congratulate the authors on a nice piece of work.

Referee #2:

I have no further comments, thank you.

2nd Confidential Review

01-Dec-2022